# Medium Chain Carboxylic Acids from Complex Organic Feedstocks by Mixed Culture Fermentation

**DOI:** 10.3390/molecules24030398

**Published:** 2019-01-22

**Authors:** Vicky De Groof, Marta Coma, Tom Arnot, David J Leak, Ana B Lanham

**Affiliations:** 1EPSRC Centre for Doctoral Training in Sustainable Chemical Technologies, University of Bath, Claverton Down, Bath BA2 7AY, UK; V.De.Groof@bath.ac.uk; 2Department of Chemical Engineering, University of Bath, Claverton Down, Bath BA2 7AY, UK; T.C.Arnot@bath.ac.uk; 3Centre for Sustainable Chemical Technologies (CSCT), University of Bath, Claverton Down, Bath BA2 7AY, UK; M.Coma@bath.ac.uk (M.C.); D.J.Leak@bath.ac.uk (D.J.L.); 4Water Innovation & Research Centre (WIRC), University of Bath, Claverton Down, Bath BA2 7AY, UK; 5Department of Biology & Biochemistry, University of Bath, Claverton Down, Bath BA2 7AY, UK

**Keywords:** anaerobic, carboxylate platform, chain elongation, circular economy, mixed microbial culture, medium chain carboxylic acid, organic waste, resource recovery, waste valorisation

## Abstract

Environmental pressures caused by population growth and consumerism require the development of resource recovery from waste, hence a circular economy approach. The production of chemicals and fuels from organic waste using mixed microbial cultures (MMC) has become promising. MMC use the synergy of bio-catalytic activities from different microorganisms to transform complex organic feedstock, such as by-products from food production and food waste. In the absence of oxygen, the feedstock can be converted into biogas through the established anaerobic digestion (AD) approach. The potential of MMC has shifted to production of intermediate AD compounds as precursors for renewable chemicals. A particular set of anaerobic pathways in MMC fermentation, known as chain elongation, can occur under specific conditions producing medium chain carboxylic acids (MCCAs) with higher value than biogas and broader applicability. This review introduces the chain elongation pathway and other bio-reactions occurring during MMC fermentation. We present an overview of the complex feedstocks used, and pinpoint the main operational parameters for MCCAs production such as temperature, pH, loading rates, inoculum, head space composition, and reactor design. The review evaluates the key findings of MCCA production using MMC, and concludes by identifying critical research targets to drive forward this promising technology as a valorisation method for complex organic waste.

## 1. Introduction

In 2016 nearly 58% of the organic fraction of municipal solid waste (OFMSW) in the EU was sent directly to landfill or incineration (estimated using the Eurostat database accessed 28/11/2018: recycling of bio-waste (cei_wm030), generation of waste by waste category (ten00108) and population on 1 January (tps00001)), resulting in undesirable environmental effects, little to no value recovery, and hence a loss of resources. However, recycling in the EU is now increasing [1], and hence separately collected organic waste is becoming more available for resource recovery or waste valorisation, i.e., the process of converting waste into energy, chemicals or materials. Technologies for bio-waste valorisation can be categorised as thermal or thermochemical such as hydrothermal liquefaction, pyrolysis and gasification, physicochemical like extraction and transesterification or biological conversion processes [2,3]. Biomass gasification has been proposed to homogenise various substrates to syngas and further process this for chemical production [4]. Reviews are available regarding technologies for waste to energy [5,6], or waste to chemicals and materials [7,8,9,10]. The choice of treatment method will depend on several factors such as type and availability of organic waste streams, e.g., the waste’s organic strength measured by chemical oxygen demand (COD) [11], relative content of biopolymers (i.e., cellulose, hemicellulose or lignin) [12], or biomass type (woody biomass, types of agricultural residues, household organic waste and sewage sludge) [13,14]. Development of a circular economy where waste is used as resource for renewable energy and chemicals will require the integration of different types of conversion processes to deal with the complexity of bio-waste and maximize resource recovery [15].

Established bio-waste valorisation technologies are composting and anaerobic digestion (AD), which each produce fertilizer and methane-rich biogas as end-products. However, the final products have relatively low economic value. For instance only € 2 worth of compost is obtained per tonne food waste [16]. AD generates a slightly more valuable product: assuming the OFMSW typically contains 306.4 g_COD_ kg^−1^ of anaerobic biodegradable content [17] and that biogas conversion yields € 0.25 worth of biogas per kg of COD [18], then a tonne of food waste will produce about € 76 worth of biogas. However, the intermediate fermentation compounds produced during AD have a higher market value.

Fermentation to accumulate the intermediate carboxylates is known as the carboxylate platform. Producing carboxylates through fermentation forms a sustainable alternative to their current production from fossil fuels or extraction in small amounts from natural oils [19]. Compared to AD, the carboxylate platform shows lower conversion yields, yet the higher product value and broader applications can result in a higher economic value [20]. In the last decade, particular interest has grown in medium chain carboxylic acids (MCCAs). They are defined as carboxylic acids with an aliphatic straight carbon chain of 6 to 12 carbon atoms, e.g., *n*-caproic acid has a straight chain of 6 carbon atoms (C6). MCCAs are more hydrophobic compared to shorter chain carboxylates, which makes them a more interesting fermentation product as it facilitates recovery from the fermentation broth [21]. In terms of potential value, C6 has a market size of 25,000 tonne per year, with an unrefined value of $1000, and refined value of $2000 to $3000 per tonne [22,23]. Overall, MCCAs have a wide range of applications: they can be applied as growth-promoting antibiotic replacements in animal feed [24,25], or be converted via various bio-, thermo-, or electro-chemical processes into bulk fuels or solvents [14,26,27,28]. The production of MCCAs as higher value products from organic waste can incentivise for improved recycling while simultaneously replacing current unsustainable production processes.

MCCAs are produced by certain bacteria in a strongly reduced anaerobic environment, via a metabolic pathway that has been recently reviewed by Spirito et al. [29]. The bacteria gain energy by combining the oxidation of an electron donor, i.e., lactic acid or ethanol, to acetyl-CoA with the reductive elongation of acetyl-CoA with acetic acid (C2), propionic acid (C3), butyric acid (C4), pentanoic acid (C5), or caproic acid (C6) generating a carboxylic acid with 2 additional carbons at each step (Figure 1). The reduction step is required to provide sufficient Gibbs free energy (ΔG) to generate ATP in the initial oxidation step, restore the NAD^+^/NADH balance in the cell, and contribute to further energy generation via electron-transport phosphorylation. Ethanol and lactic acid have similar thermodynamic capacity to act as electron donors [30]. This chain elongation pathway is called the reverse β-oxidation pathway, since it is seen as the reversed biochemical degradation or β-oxidation of fatty acids.

Instead of using pure or engineered cultures, a consortium of microorganisms has more potential to deal with complex and variable feedstock such as organic waste. Mixed microbial cultures (MMC), also referred to as microbiomes, are communities of microorganisms within a well-defined environment of specific physicochemical properties [31]. Microbiomes are employed in biotechnology, for example in anaerobic digestion (AD), and in bioremediation by cultivating communities within contaminated soils [32,33]. The term “microbiome” is used to describe the mixed microbial communities related to the human and animal gut, mouth or skin, or plant rhizospheres.

The first report of MMC that produced MCCAs dates back to the mid-19th Century, where Béchamp attributed the production of approx. 6 g_COD_ L^−1^ C6 from ethanol, meat extract and chalk in a fermentation reactor to microbial activity [34]. A few decades later in the early 20th Century, an oily, immiscible layer comprising 5.3 g_COD_ L^−1^ C4 and 6.4 g_COD_ L^−1^ C6 was produced in a 30-day fermentation with impure cultures from a nutrient medium containing 24 g_COD_ L^−1^ ethanol [35]. Further microscopic study of the fermentation sludge revealed a consortia of microorganisms comprising methanogenic archaea and spore-forming bacteria [35]. By contrast with pure cultures, MMC do not require sterilisation, can degrade a complex feedstock, show a resilience to operational upsets [36], and allow continuous, long-term operation [37]. These advantages provide a strong argument for utilising microbiomes.

In MMC, conversion of organic substrates occurs following a cascade of steps catalysed by different microorganisms that form synergistic and competitive interactions, resulting in a complex microbial ecosystem with a versatile metabolic capacity [38]. The different microbial groups can convert organic molecules into substrates available for chain elongating bacteria. In general, biodegradable organics are hydrolysed and fermented to intermediate compounds that acidify the medium, i.e., acidogenesis, including hydrogen gas (H_2_), lactic acid, ethanol, formic acid (C1) and volatile fatty acids (VFAs), i.e., straight short chain carboxylic acids with 2 to 4 carbon atoms. The accumulated intermediates can undergo several secondary bioconversion steps, including chain elongation to produce MCCAs (Figure 2) [26]. For instance, co-culture of the chain elongating bacteria *Clostridium kluyveri* with specific cellulolytic species or a rumen microbiome showed chain elongation potential from a cellulose substrate and ethanol [39,40]. The supporting community can even be designed or selected to allow chain elongation from a specific compound, such as glycerol or syngas (CO) [41,42,43], or allow the use of alternative electron donors such as, for instance, the cathode in a bio-electrochemical system [44,45].

While it is generally believed that specific operational conditions allow development of a MMC for a functional and stable process [46], the broad metabolic capacity also gives rise to a set of various competitive reactions and by-products, especially when utilising a complex feedstock. Manipulating the environmental conditions, by regulating operation, allows some control to be exerted on the product spectrum, as it affects the thermodynamics of conversion processes, and therefore the microbiome composition that catalyses these conversions. However, current knowledge of control over the product outcome to improve MCCA yields in MMC fermentation is limited since experiments that use complex feedstock for MCCAs production have only emerged in the past few years.

While the operational conditions that select for other MMC fermentation products such as volatile fatty acids (VFAs) [47] and hydrogen (H_2_) [48] have been reviewed, the operational conditions or process set-up that allow MCC to be steered towards MCCA formation have to be further evaluated. A recent review is available regarding the use of bio-electrochemical systems for MCCA production as a complementary technology to AD [49]. Certain other reviews include a section on MCCAs as potential MMC fermentation products, either in the context of operational control applied in AD [50], or the contexts of a biorefinery [51], wastewater treatment [11] or food waste treatment [21,52,53,54]. However, a focussed analysis of the literature to identify and connect key operational parameters to target MCCA production from MMC fermentation of complex feedstocks is lacking. Therefore, this work aims to analyse the current literature, and hence complement existing reviews. For this, studies were included that specifically target chain elongation, but the scope was extended to include other MMC-based studies that have noted MCCA as by-products from, for instance, VFA or H_2_ production. Concentrations and production rates are converted to a COD-basis to allow comparison between studies using different reporting concentrations (Appendix A). The review evaluates the key operational parameters for MCCA production from complex substrates using MMC, with the objective of stimulating and accelerating research to produce sustainable, bio-based fuels and chemicals from organic waste. In addition, a database was generated from the experimental data available in the literature regarding MCCA production using MMC fermentation [55].

## 2. Chain Elongation Behaviour of Pure Cultures Can Be Extended for MMC 

Chain elongation via ethanol is the most studied pathway to date. The mechanism has been elucidated by studying *Clostridium kluyveri*, a gram-positive, spore-forming bacteria from the phylum of Firmicutes whose whole genome has been published [56]. For each molecule of ethanol oxidized to C2, resulting in substrate-level ATP-generation and production of H_2_, five molecules of ethanol enter the reverse β-oxidation pathway as acetyl-CoA and elongate five molecules of C2 to C4. Subsequently, C4 can be elongated to C6 via ethanol-derived acetyl-CoA addition (Table 1, Equations (1) and (2)) [29]. In reality, the pathway of *C. kluyveri* has a more flexible stoichiometry influenced by substrate concentrations, ratio of ethanol to acetate, and the partial pressure of H_2_ (Table 1, Equations (3)–(6)) [57,58,59]. It also has a broader substrate range including propanol as an electron donor, or propionate (C3), succinate, malonate, 3-butenoate, 4-hydroxybutyrate and crotonate as electron acceptors [39,60,61]. Pure culture fermentations of *C. kluyveri* fed with ethanol and C2 mixtures have been reported to produce C6 up to 10.2 g_COD_ L^−1^ d^−1^ in continuous culture [62] and to reach concentrations up to 30.7 g_COD_ L^−1^ after 72 h of batch culture [60].

Chain elongation via lactic acid has been reported for other bacteria in the phylum of Firmicutes, such as *Megasphaera elsdenii* [63] and a *Ruminococcaceae* bacterium CPB6 [64]. Other wild-type bacteria are known to perform chain elongation and produce C6 (and C8) using more “exotic” chain elongating substrates such as simple sugars, polyols, methanol, amino acids and H_2_ and CO_2_ gas mixtures as reviewed by Angenent et al. [58]. In addition, pathways have been engineered to produce C6. For instance, to improve yields the genes from *Megasphaera* sp. were expressed in *Escherichia coli*, and approx. 1.17 g_COD_ L^−1^ d^−1^ C6 was obtained after 36 h of batch fermentation [65]. To develop a more thermo-tolerant and acid-resistant biocatalyst, biosynthetic pathways have been constructed in the yeast *Kluyveromyces marxianus* [66]. Single-strains or engineered cultures have their place when the product is of high enough value and require a certain purity. Overall, the production of MCCAs, and other medium chain chemicals, using pure, engineered cultures has been recently reviewed by other authors, e.g., Sarria et al. [67] and Su et al. [68].

However, when it comes to breaking down a complex feedstock such as organic waste, the focus of this review, pure cultures have limited metabolic capacity, reducing their potential for an effective treatment and requiring more expensive processing such as media sterilisation [71]. This can be circumvented by using MMC instead of pure cultures. Chain elongation in MMC happens in a similar manner than with pure cultures. For instance, microbiomes grown in ethanol-rich conditions show similar characteristics to pure culture fermentation, such as higher specificity towards longer chain carboxylates at higher ethanol/acetate ratios [72,73] and elongation towards a mixture of even- and uneven MCCA in the presence of propanol or C3 [74,75]. It should be noted that MMC are unable to use either 4-carbon alcohols or 5-carbon carboxylates as initial substrate sources for chain elongation at similar concentrations than for example ethanol or acetate [75]. This may be due to longer chain substrates having higher toxicity and possible inhibition of the microbiome. In addition, microbiomes are capable of adapting to substrate fluctuations: MMC obtained from ethanol-based chain elongation reactors acclimatised to produce C6 when fed with methanol or lactic acid as an alternate electron donor [76,77].

## 3. Thermodynamic Models and MMC Composition Determine Competitive Processes 

Successful production of MCCAs requires elimination of competing reactions that could consume the substrate or product. Some example reactions include methanogenesis, sulphate reduction, lactate reduction to propionate (C3), excessive oxidation of ethanol, and reduction or oxidation of carboxylic acids, as described in Table 2. Since anaerobic ecosystems are energy-limited with ΔG of conversion processes being close to 0 kJ mol^−1^ (Table 1 and Table 2) [78], the thermodynamic favourability of bioconversion processes can shift by small changes in substrate or product concentrations, pH and temperature, partial pressure of gases in reactor headspace, or substrate availability [50,79,80]. The resulting thermodynamic constraints select for the viable bioconversion reactions and, hence, the composition of microbiome that has the most efficient catabolic system [81,82]. Therefore, strategies to inhibit competitive reactions can be classified as; (i) the inhibition of a specific, competitive trophic group, or (ii) the engineering of the fermentation environment to reduce the potential competitive reactions. For instance, methanogenesis, the ability to produce CH_4_, is limited to certain archaea. Since CH_4_ has the lowest free energy content per electron upon oxidation to CO_2_ under anaerobic conditions, and automatically leaves the reactor as a gas, it will be produced by methanogens in MMC to optimally use the energy available [71]. To ensure that C2 or H_2_ are not lost to CH_4_ and CO_2_ (Table 2, Equations (10) and (11)), specific methanogenic inhibitors can be added to promote chain elongation. For instance, in batch fermentation of a synthetic substrate containing ethanol and C2, the addition of 2-bromoethylsulfonate (BES) tripled C6 production to 19 g_COD_ L^−1^ [83]. Alternatively, to avoid the cost of such chemicals, specific operational conditions such as pH or hydraulic retention time (HRT) can be selected to inhibit methanogens as discussed subsequently. Another unwanted trophic group are the sulphate reducing bacteria. A sulphur-rich feedstock will result in sulphate reduction, as this is more thermodynamically favourable than C6 production (Table 2, Equation (12)), generating sulphide, which is both toxic for most bacteria and corrosive to fermentation equipment [84].

Thermodynamic models are useful tools to improve understanding of the chain elongation pathway in MMC and to determine which operational parameters allow to regulate the product spectrum. Research has developed kinetic and thermodynamic models based on pure culture chain elongation using *C. kluyveri* [69,85] or metabolic energy-based models to predict MMC fermentation of simple substrates such as glucose [86]. Such models can help understanding the occurrence of chain elongation at different ethanol concentrations [58], or at varying H_2_:CO_2_ ratios [30]. There is a lack of models that evaluate the thermodynamics of the lactic acid-based chain elongation route. At standard conditions lactate reduction (Table 2, Equations (13) and (14)) releases more energy than chain elongation via lactic acid (Table 1, Equation (9)). Experimentally, Kucek et al. [77] found increasing lactate loading rate with a synthetic feedstock initially improved chain elongation in MMC fermentation, yet increasing influent lactic acid from 9.1 to 16.2 g_COD_ L^−1^ d^−1^ led to a collapse of C6 productivity to 3.0 g_COD_ L^−1^ d^−1^ while C3 production increased to 5.5 g_COD_ L^−1^ d^−1^. This was attributed to the competitive acrylate pathway being stimulated ahead of chain elongation at elevated lactic acid concentrations [77,87]. In contrast, another study operating with an excess of lactic acid did not report C3 production; the addition of three spikes in a fed batch-style adding a total about 26.7 g_COD_ L^−1^ lactic acid to the synthetic medium resulted in C6 accumulation of up to 51.7 g_COD_ L^−1^ [88]. The development of thermodynamic models focusing on lactic acid-based chain elongation might shed more light on these competitive pathways.

Modelling thermodynamics only goes so far, and the composition of the microbiome must be considered as this can influence the microbiome’s metabolic capacity. The results from microbial community composition analysis using 16s rRNA gene sequencing of the mentioned studies cited above indicate they had a different microbiome structure. The fermentation where the acrylate pathway took over, had a wider variety of prokaryotic families and was dominated by *Acinetobacter* spp. (approx. 60% relative abundance) and the operational taxonomic units belonging to *Ruminococcaceae* were less than 10% [77]. On the contrary, the study with minimal C3 production was dominated by a Clostridium cluster IV group (79.1%, belonging to *Ruminococcaceae*) [88].

Studying the microbiome composition improves the understanding of the MMC fermentation mechanisms. For instance, when following the microbial community dynamics of maize silage fermentation in a leach bed reactor (LBR), Sträuber et al. [89] found *Lactobacillus* and *Acetobacter* strains dominated during the first days of operation, with lactic and acetic acid as concurrent products. However, *Clostridium* species became dominant on Days 3 and 4 resulting in a pH increase and C4 and C6 production, and in turn these were overgrown during Days 5 to 7 by other phylotypes capable of using more complex polysaccharides by different metabolism [89]. Further investigation of microbial interactions and synergies will allow better design of MCCA production processes from complex feedstocks, for example operating in sequential batch mode to allow the different trophic groups to first accumulate ethanol, lactic acid, H_2_ or VFAs for subsequent chain elongation. Only 20 studies on MCCA production could be found, so far, that include an analysis of the microbial community. This usually involves DNA extraction and sequencing of 16s rRNA amplicon and comparison to sequence databases [23,70,73,75,77,83,88,90,91,92,93,94,95,96,97,98,99], sometimes in addition to other community analysis such as flow cytometry [100], analysis of terminal restriction fragment length polymorphisms (T-RFLP) [101] or microscopic evaluation [102].

Recently, Scarborough et al. [103] combined metagenomic, metatranscriptomic and thermodynamic analysis of samples from a reactor microbiome fermenting a lignocellulosic-based feedstock in continuous stirred-tank reactor (CSTR) mode, which allowed, for instance, affiliation of *Lactobacillus* and members of the *Coriobacteriaceae* family to hydrolysis and primary fermentation, and organisms related to *Lachnospiraceae* and *Eubacteriaceae* to MCCA production. In addition, the recent advancements in metagenomic and metatranscriptomic analysis led to the proposition that other MCCA-producing pathways occur in a microbiome, such as the fatty acid biosynthesis pathway, alongside the reverse β-oxidation pathway [104].

Research is necessary to expand the thermodynamic models to include both the MMC composition, which indicates the potential bio-reactions in a system, and the composition of complex feedstocks. The development of these models will complement the understanding obtained from experimental studies, and will help in determining the operational parameters which select for MCCA production over competitive reactions. In addition, culture-independent analysis and increased application of “omics” approaches on MMC fermentation studies will be essential to enhance our understanding of the underlying mechanisms that include competitive and synergistic processes and the importance of the MMC composition.

## 4. Bio-Waste Composition and Its Effect on Chain Elongation

A feedstock suitable for chain elongation should provide the necessary substrates, i.e., VFAs and electron donors such as ethanol or lactic acid. Chain elongation substrates can either be directly present in the feedstock, indirectly produced from primary fermentation in vivo, or supplemented. The highest MCCA production rates obtained in MMC fermentation used a synthetic feedstock, hence a readily bio-available substrate. In up-flow reactors (URs) with biomass retention, 115.2 g_COD_ L^−1^ d^−1^ for C6 [105] and 19.4 g_COD_ L^−1^ d^−1^ for C8 [96] were obtained from ethanol and C2 mixtures. These rates are more than 10 times higher than that achieved so far using complex, un-supplemented feedstocks (Table 3). If electron donors, such as ethanol or lactic acid, are supplemented, selectivity of secondary fermentation is enhanced towards chain elongation. Ethanol-supplemented organic waste streams have reached production rates that lie somewhat in between synthetic and complex feedstocks. The maximum reported is 60.7 g_COD_ L^−1^ d^−1^ C6 and 2.13 g_COD_ L^−1^ d^−1^ C8 for pre-fermented OFMSW supplemented with 97.4 g_COD_ L^−1^ d^−1^ ethanol [106]. Supplementation of 21.3 g_COD_ L^−1^ lactic acid to pre-treated grass in batch fermentation with an adapted inoculum resulted in a total C6 concentration of 24.1 g_COD_ L^−1^ after 1 day [95].

When applying a supplementation strategy, certain experiments show that excessive concentrations of ethanol and lactic acid should be avoided. An upper limit for ethanol-based chain elongation is reported at 97.4 g_COD_ L^−1^ after which it exerts an inhibitory effect [107]. Grootscholten et al. [108] spiked a LBR processing OFMSW at four intervals with 11.2 g_COD_ L^−1^ ethanol during batch fermentation. This increased MCCA concentration from 4.0 to 6.0 g_COD_ L^−1^ for C6 and 0.0 to 1.2 g_COD_ L^−1^ for C8 compared to a non-supplemented control experiment, yet the total amount of carboxylic acids produced was lower [108]. This was attributed to ethanol inhibition of hydrolytic and acidogenic bacteria [108]. To avoid ethanol inhibition limiting primary fermentation, Grootscholten et al. [106] suggested the use of a two-stage system where hydrolysis and primary fermentation of OFMSW occurs in the first batch phase, and in the second the pre-fermented OFMSW is supplemented with ethanol to select for chain elongation in a CSTR. The disadvantages of two-stage systems are the increased operational complexity and additional capital and operational costs.

A life-cycle assessment on C6 production from ethanol-supplemented food waste fermentation in a lab- and pilot-scale system revealed that the largest environmental effects (acidification and eutrophication potential) result from addition of caustic soda to control pH and ethanol as electron donor [109]. Thus, supplementation of electron donors to stimulate chain elongation should be minimized or avoided. Instead of supplementing the feedstock, several studies have split the overall fermentation into two stages. Firstly, specific operational conditions are selected to accumulate ethanol or lactic acid. Then in a second phase, the leftover organics in the effluent are fermented towards VFAs and elongated with the electron donor under chain elongating conditions. Ethanol-rich substrates fed into chain elongation reactors have been obtained from yeast-based fermentations, such as for the production of bio-ethanol which generates an ethanol-rich beer and a residue after distillation called stillage [27,37,93], or residues from the production of wine [97], or effluent from syngas fermentation [110]. Lactate-rich substrates can be obtained via MMC fermentations selective for lactic acid, such as effluent from thermophilic acid whey fermentation [94], and pre-fermented grass (i.e., grass silage) [95] or maize silage [27,89]. Whey is rich in lactose, simple sugars such as fructose that are easily fermented to produce lactic acid [111]. Food waste contains lactic acid bacteria that are easily enriched in MMC fermentation, to produce lactic acid up to concentrations of 21.3 to 48 g_COD_ L^−1^ [112,113,114]. Maximum production using ethanol- and lactic acid-rich streams in MMC fermentation without additional supplementation have been reached using diluted beer (7.52 g_COD_ L^−1^ d^−1^ C6) [46] and acidified whey from the quark industry (5.12 g_COD_ L^−1^ d^−1^ C6) [85]. The disadvantages of two-stage systems for accumulation of electron donors are the increased operational complexity and additional capital and operational costs.

An alternative to supplemented or pre-fermented organic feedstock, would be to operate a single-phase process where the substrate itself is converted to lactic acid and/or ethanol in parallel with the chain elongation reactions. Recently, C6 has been produced in a one-stage LBR from food waste up to concentrations of 21.8 g_COD_ L^−1^ and a rate of 3.12 g_COD_ L^−1^ d^−1^ [91], comparable to a supplemented and/or two-stage system. In this study, batch tests inoculated with the LBR leachate showed that lactic acid was consumed in favour of ethanol as the electron donor [91]. Other studies aimed at VFA or H_2_ production have produced C6 in a single-stage approach from food waste, without supplementation, and at similar production rates. In studies aimed at H_2_ production using glucose-rich synthetic wastewater [115] or simulated food waste [116] as feedstock, C6 was produced at rates higher than experiments which targeted MCCA production (Table 3). Therefore, food waste and similar complex substrates might be a promising feedstock.

The use of complex feedstock comes with a set of challenges. Firstly, the composition of the feedstock will influence certain competitive processes and the intermediates that result from hydrolysis and acidogenesis, and therefore affects the potential for chain elongation. For example, VFA yields in MMC fermentation have shown to be less for lipid-rich waste compared to carbohydrate- or protein-rich wastes, due to difficulties in hydrolysis [117]. In addition, the type of carbohydrates [118] or proteins [119] in the substrate also influences the metabolic pathways. Secondly, it is important to recognise that seasonal and geographical variation in organic waste streams impacts composition and availability of feedstock. This can often become a challenge when designing a suitable treatment or valorisation (bio-)process that is robust and flexible, especially when relying on pure microbial cultures. For instance, OFMSW may contain small quantities of bio-waste from various origins, such as discarded oils, fruits, animal-derived products and more lignocellulose-rich leaves and stems from vegetables, resulting in an overall feedstock comprising both easily biodegradable carbohydrates and proteins as more recalcitrant matter [120]. It was found that a LBR fed with OFMSW which was richer in green waste had lower production of carboxylic acids compared to OFMSW containing more food waste [121]. Lastly, not every type of substrate will allow chain elongation to occur. In batch experiments aimed at VFA production, seven types of “food waste” feedstocks were tested and only four produced C6. These were cheese whey, sugarcane molasses, the organic fraction of municipal solid waste, and winery effluent [122]. Similarly, in sequential batch reactors (SBRs), the fermentation of cheese whey resulted in 0.12 g_COD_ L^−1^ d^−1^ C6, while in similar operating conditions the fermentation of sewage sludge produced more total carboxylic acids but very little C6 [123]. Therefore, the right type of complex feedstock should be selected for MCCA production to be successful—this is not well understood and requires further investigation.

## 5. Environmental Conditions that Influence Chain Elongation

### 5.1. Mesophilic Temperatures Seem to Benefit Chain Elongation

Temperature has a significant influence on energy released from reactions [80], alters the microbial community composition [129], and affects the kinetic rates of metabolic reactions. The amount of research focusing on the effect of temperature on chain elongation is limited, yet results so far indicate operation in the mesophilic range, typically between 30 °C and 45 °C, is preferred over thermophilic conditions (Table 4). In a long term experiment fermenting ethanol-rich beer (from a yeast fermentation) in a SBR with in-line product extraction, selectivity for C6 nearly tripled after reducing the temperature from 55 to 30 °C [130].

The effect of temperature on acidogenesis and VFA production has been studied in more depth and could provide further insight. In a study evaluating acidogenesis of cafeteria food waste, trials at 25, 35 and 45 °C gave the highest overall carboxylic acid yield at 35 °C (~23.5 g L^−1^ total carboxylic acids with ~3.1% C6), yet the C6 yield was higher at 45 °C (~19.8 g L^−1^ total carboxylic acids with 24% C6) [125]. VFA-targeted fermentation of pre-treated cellulosic biomass with meso- (40 °C) or thermophilic (55 °C) operation showed a significant divergence in the proteins and enzymes present and in MMC composition. Mesophilic operation allowed C6 formation to occur (3 to 4% of carboxylic acids) while no C6 was found under thermophilic conditions [129,131]. The thermophilic reactor maintained a community similar to the inoculum, rich in Clostridia, while the mesophilic system showed a wider variety of taxa which was dominated by members in the Bacteroidetes phylum [129]. While there are some indications that a mesophilic temperature favours chain elongation, more evidence is needed to understand the processes that affect chain elongation productivity at different temperatures. These could be attributed to changes in the collective microbiome metabolism and/or composition, as well as to thermodynamic and kinetic shifts to competing reactions.

### 5.2. The Influence of pH and Buffer Capacity on MCCA Production

Metabolic pathways, hydrolysis, thermodynamics and, product outcome are significantly influenced by pH in MMC fermentation [123]. For example, in fermentation of cheese whey poor control of pH led to fluctuating carboxylic acid concentrations and hindered establishment of steady state [111]. The specific effects of pH on chain elongation are not straightforward and various factors must be considered.

Firstly, pH determines the dissociation/association equilibrium between carboxylic acids and carboxylates, and therefore the impact of product toxicity due to the acidic form. Secondly, pH determines the CO_2_ and carbonate distributions, which affects buffer capacity, and its availability to certain organisms. For instance *C. kluyveri* requires incorporation of CO_2_ in its biomass [132]. Thirdly, thermodynamic feasibility depends on pH. Indeed, chain elongation is less thermodynamically favourable under alkaline conditions [133]. Lastly, the effect on the microbiome composition and microbial competition has to be taken into account. For instance, the highest reported production rate of C6 and C8 was obtained at neutral pH, yet a specific methanogenic inhibitor had to be added since acetoclastic methanogens compete under these conditions, and consume C2 for CH_4_ production, reducing its availability for chain elongation [105]. Methanogenesis occurs within a pH range of 6.0–8.3 while acidogenic bacteria can tolerate lower pH [134]. Thus a strategy to minimize CH_4_ production is to operate at slight acidic pH (5.0–5.5), and this has been applied in various VFA or MCCA producing experiments [100,123,130].

The effects of pH on primary fermentation also have to be considered. Experimental studies have shown that controlling pH at alkaline or neutral values improved liquefaction and hydrolysis of solids in organic waste, but it did not support chain elongation since short chain carboxylates were accumulated without production of MCCAs [126,135,136,137]. In addition, it is necessary to consider the cost of pH correction, as for example, food waste has an average pH of 5.1 [138]. Operating at neutral or alkaline pH would require substantial addition of pH-correcting chemicals, and hence lower the economic feasibility [89].

Despite the complex effects of pH on MMC fermentation, some general effects on product outcome are reported for organic or synthetic waste fermentation. A shift to lactic acid or ethanol production was observed in two separate studies, one at pH < 3.6 [115] and another at pH < 4.5 [139]. At slightly higher pH range of 5.0 to 5.5 C3 accumulated as main fermentation product [139]. Two studies have investigated the effect of pH on H_2_ production, an important element necessary for chain elongation, which found different results: one suggested optimal production around a pH of 4 [115] and another at pH of 8 [140]. Finally, MCCA production can occur under slightly acidic conditions (5 < pH < 6), although isolates of MCCA producers such as *C. kluyveri* operate best at near-neutral pH [106]. Therefore in order to favour chain elongation the evidence so far suggests that whilst the target microbial communities might operate across a broader pH range, a slightly acidic or neutral pH might result in the best compromise between thermodynamics, kinetics and microbial competition. The pH might need optimisation depending on other operational parameters and fermentation characteristics, such as other factors influencing competitive reactions or substrate composition.

### 5.3. The Influence of pH on Product Toxicity

Another aspect linked with operational pH is product toxicity from C6 and C8 compounds. Carboxylic acids up to a carbon chain length of 18 are potent antimicrobial compounds that interact with lipid bilayers of cell membranes, disrupting the energy generating mechanisms, internal pH, and cell integrity [141,142]. At acidic pH a higher percentage of MCCAs are present in their un-dissociated, toxic acid form. With a pKa of 4.85 for C6 and 4.89 for C8, 50% of the total carboxylate is found in the acidic form at a pH equal to the pKa. C6 and C8 showed similar toxicity in *E. coli* at 10.2 to 14.1 g_COD_ L^−1^ at pH 7 [143]. For mixed culture chain elongation, different inhibitory concentrations have been reported. Angenent et al. [58] observed the inhibitory concentration of protonated C6 to be 1.92 g_COD_ L^−1^ in a system using yeast-fermentation beer. Khor et al. [95] obtained stable C6 production around pH 5.5 at just below this limit during fermentation of grass silage with adapted MMC; however, supplementation of lactic acid led to accumulation of protonated C6 concentrations up until 4.4 g_COD_ L^−1^, with total C6 concentrations close to the solubility limit of C6 (≈22.7 g_COD_ L^−1^).

### 5.4. Buffering Capacity and Potential of Self-Regulation of pH

The buffering capacity or alkalinity of the fermentation broth is the capacity to neutralize acids via the presence of compounds such as phosphate, bicarbonate, carbonate and hydroxides. If there is limited buffer capacity or no continuous pH control, the pH in MMC fermentation quickly drops to minimum of 3.0 due to acidification [140,149]. Fermentation of Brewer’s Spent Grain was shown to reduce pH from 6.5 to 3.8 after only 1 day due to lactic acid production [133]. Also ammonium (NH_4_^+^) released during hydrolysis of nitrogen-rich feedstock neutralises acids during fermentation by anaerobic consortia [150]. Sometimes chain elongation experiments include a buffer in their synthetic medium [127]; alternatively bicarbonate is added as nutrient requirement for *C. kluyveri* [72,74,110].

No studies could be found that evaluate the effect of alkalinity on chain elongation. However, for other types of MMC fermentation alkalinity and buffering mechanisms have been studied (Table 4). Examples include AD, H_2_ production or acidogenesis for VFA production, which sometimes include the reporting of C6. Alkalinity in MMC fermentation in AD is well studied and its measurement serves as a tool to indicate early warning of unwanted acidification and pH destabilisation [151]. The optimal value of alkalinity usually lies between 1.0–5.0 g_CaCO__3_ L^−1^ [144,145]. For H_2_ production by MMC from organic waste, an alkalinity optimum was found around 0.11 g_CaCO__3_ g_COD_^−1^ with excessive alkalinity resulting in increased osmotic pressure [146]. Improved VFA production has been noted for fermentation reactors processing cheese whey, the organic fraction of municipal solid waste or synthetic soft drink wastewater with higher alkalinity ranging from 2.4 to above 30 g_CaCO_3__ L^−1^ [122,147,148]. Comparison of buffered (pH ≈ 7) and unbuffered (pH drop to approx. 6) batch fermentation of vegetable and salad waste over 70 days showed that addition of ~13.3 g L^−1^ NaHCO_3_ tripled C6 concentrations for the first 52 days to a maximum of 6.6 g_COD_ L^−1^. However, over longer operating times production decreased in the buffered system due to methanogenesis resulting in similar C6 yield for the two systems at the end of operation [126]. Therefore, there are indications that a minimum alkalinity would benefit MCCA production, but this needs to be further investigated.

Studies aimed at VFA production from complex substrates indicate a self-regulation of pH in MMC fermentation in the range of 5.2 to 6.7 [135]. A similar pH stabilisation effect has been noted by studies directed at MCCA formation. For example, batch fermentation of food waste generated a pH drop from 7 to ~5.3 due to hydrolysis and primary fermentation; after 20 days it rose again to pH 5.8 with production of MCCAs [90]. For continuous processes this effect results in an overall pH stabilisation. A stable pH between 5.5 and 6.2 occured while feeding a lactic-rich substrate at pH 4.5 to 5 [95]. Chain elongation via lactic acid consumes protons (Table 1, Equations (7)–(9)), although no studies have investigated the link between lactic acid-mediated elongation and the buffering of pH. Further studies are needed to show the minimal alkalinity required for chain elongation in MMC to sustain a minimum buffer capacity, which also allows continuous operation at optimal pH, and with limited or no pH control.

## 6. The Push towards Chain Elongation by Organic Overloading

For AD applications, organic overloading is defined as the COD loading rate exceeding the degradation capacity of the anaerobic microbiome, leading to accumulation of VFAs and hence a decrease of pH which inhibits methanogens [152]. For VFA and MCCA production, organic overloading is deliberately employed to promote carboxylic acid accumulation and limit competitive processes. Elevated carboxylic acid concentrations inhibit methanogens, as found, for instance, for C4 where only ~2.4 g_COD_ L^−1^ at pH 6 was shown to inhibit 90% of methanogens in a thermophilic batch fermentation [130].

The organic load at start-up can be represented by the food-to-microorganisms ratio (F/M). This is defined as the amount of feedstock introduced, expressed as COD or volatile solids (VS), relative to the amount of biomass, estimated as VS or volatile suspended solids (VSS) in the inoculum [9,153]. In anaerobic consortia fermenting food waste, organic overloading and VFA accumulation usually occur at start-up with a F/M ratio > 1 gCOD_fed_ gVS_inoculum_^−1^ and carboxylates accumulate at an optimal F/M ratio of 5 gCOD_fed_ gVS_inoculum_^−1^ [9]. As with the acidogenic fermentation of synthetic soft drink wastewater, a F/M ratio of 4.0 gCOD_fed_ gVSS_inoculum_^−1^ was found to be optimum for C6 production compared to F/M ratios of either 1.6 and 6.4 gCOD_fed_ gVSS_inoculum_^−1^ [148].

A positive relationship is found for MCCA production and higher organic loading rates (OLRs) using synthetic or supplemented substrates and pure cultures or MMC [96]. However, for complex feedstock, whilst the same mechanism would be expected, the relationship between OLR and MCCA production is less straightforward (Figure 3). When using a synthetic medium or supplementation of electron donors the OLR, expressed in terms of total COD, is directly related to the amount of bio-available substrate. On the contrary, when using complex feedstock, the OLR does not necessarily indicate the presence of chain elongation substrates or anaerobic biodegradable content, which can be converted to chain elongation compounds. Namely, fractions can be present in complex feedstocks that can be chemically oxidised, and thus contribute to the total COD, but are not easily biologically degraded. For instance, bio-waste collected from municipalities had a higher anaerobic biodegradability (90.8 ± 3.7%) compared to food waste collected from a vegetarian restaurant (66.9 ± 6.4%) even though their total COD values were within a similar range, i.e., 337 ± 14 g_COD_ kg_WW_^−1^ and 303 ± 18 g_COD_ kg_WW_^−1^ respectively [17]. MMC fermentation experiments for MCCA products using complex feedstocks have only recently been conducted, and little is known regarding the need for easily biodegradable feedstock in order to exert sufficient OLR to accumulate chain elongation substrates for C6 production. In AD, high-COD substrates such as food waste or food processing by-products result in problematic increased retention times and low throughput, to prevent organic overloading and hence such feedstocks show greater potential for production of MCCAs rather than biogas.

## 7. Effect of Inoculum on Achieving MCCA

Selection of the appropriate inoculum in biotechnological applications is critical to ensure catalysis of the required bio-conversion process. Cultures used for inoculating reactor experiments with complex substrates for carboxylate production contain a large variety of microorganisms (high richness) and include anaerobic sludges from AD [147] or wastewater treatment [133], marine sediments [107,129], rumen samples [30,40], mixtures of cultures [155] or microbiomes enriched in chain elongators obtained from lab-scale reactors [83,91,96]. For production of VFAs, H_2_ or other MMC fermentation products, the inoculum can be physico-chemically pretreated in order to suppress methanogenesis, e.g., heat shock and/or acid/alkali conditioning [156,157]. However, no studies could be found on the effect of inoculum pretreatment on MCCA production specifically. Studies on MMC fermentation towards H_2_ or VFAs have shown that the microbial composition was altered and the MMC fermentation product profile shifted depending on inoculum pretreatment method applied [7,47,158,159]. For chain elongation in particular, Cavalcante et al. [69] has discussed thermal pretreatment as a potential selective pressure since various chain elongating bacteria have been allocated to the genus *Clostridium*, that due to its spore-forming abilities could be selected for by heat shock. The limited knowledge regarding pretreatment of inoculum to select for chain elongation would be worthwhile to further investigate.

Separate inoculation is not always required. Some experiments showed C6 production from organic waste by simply using the endogenous MMC present in the feedstock [108,135]. In addition, regardless of the initial inoculum source (i.e., sludge from full-scale AD plant or lab-scale fermentation reactors) microbiomes grown on synthetic substrates with ethanol as the main electron donor are enriched by species closely related to *C. kluyveri* due to adaptation over time [70,101].

As shown in Figure 4, for batch experiments, the most commonly used inoculum types are AD sludge or enriched microbiomes, i.e., previously enriched and adapted to chain elongation conditions in the lab. Yields seem to be similar for both types of inoculum, perhaps due to high microbial richness. However, the total MCCA concentration accumulated with an AD inoculum had a 5 times lower upper range than when an adapted microbiome is used, indicating that tolerance towards toxic MCCA concentrations can be developed (Figure 4).

This tolerance to higher C6 concentrations from acclimatised microbiomes has been recently reported using batch inhibition assays with synthetic media and differently cultured sludges [160]. The same adaptability hypothesis was used to explain productions of longer MCCA such as C8 [92]. However, batch experiments inoculated with an enriched microbiome are often operated at a higher organic load that could also enhance MCCA accumulation, and these parameters are inter-connected. From the analysed studies it also appears that bio-augmentation does not improve considerably the MCCA yields, but it does impact on the product concentration with more MCCAs produced. It has been shown that bio-augmentation with *C. kluyveri* improves yields and even results in chain elongation up to decanoic acid (C10) [90]. From the data currently available, inoculum selection could contribute to MCCA specificity, yet further investigations are needed to evaluate the importance of inoculum selection.

## 8. The Influence of Partial Pressures in Reactor Headspace

### 8.1. The Aerotolerance of MMC Fermentation

*M. elsdenii* and *C. kluyveri*, two well-characterised bacteria capable of chain elongation, are both strictly anaerobic [61,132]. Anaerobic conditions are easily maintained in laboratory experiments by flushing the headspace with N_2_, CO_2_ or H_2_ or mixtures thereof. However, full-scale or pilot reactors are not always operated fully anaerobically [164]. Therefore, it is important to understand the sensitivity to the presence of oxygen on MMC when producing MMCAs.

In a study investigating the aerotolerance of MMC fermentation of shredded paper and chicken manure, intermittent air exposure had no significant influence on bacterial community composition, however, it did select for shorter chain carboxylates, whilst stricter anaerobic conditions improved chain elongation [164]. On the contrary, in another study using different types of pre-treated corn fibre, air exposure did not lower the C6 production rate [36]. There is still very little evidence of the capacity of MMC cultures performing chain elongation in regards to their tolerance of oxygen and its impact on the production of MMCAs. This should be the subject of further studies as it is an important design parameter for experimental and full-scale chain elongation processes.

### 8.2. Minimal and Balanced CO_2_ and H_2_ Required for Chain Elongation

A sufficiently high H_2_ partial pressure (*p*H_2_) in the headspace gas is an important parameter to limit the impact of competitive processes and is seen as a central strategy to ensure reductive conditions for reverse β-oxidation [130]. However, the mechanism of how *p*H_2_ affects chain elongation is not fully understood, and various values have been reported (Table 5). Similarly, *p*H_2_ controls the VFA yielding of anaerobic MMC fermentation such as AD. Different *p*H_2_ have been reported to inhibit or thermodynamically constrain certain bioconversion processes thereby influencing the VFA product spectrum [165,166].

In thermodynamic fermentation models, it is assumed that dissolved H_2_ affects the NADH/NAD^+^ ratio directly, and hence the thermodynamic feasibility of certain pathways [81]. Nevertheless, experimental work could not find a direct correlation between the two, thus possibly indicating a more complex effect where alternative electron carriers (e.g., ferrodoxin) are involved in MMC [167]. H_2_ is a product from ethanol- and lactic acid oxidation occurring during chain elongation [29], yet it is also an indirect electron donor for chain elongation by its capability of reducing C2 to ethanol [127]. A minimum *p*H_2_ (a composition > 0.007% at standard conditions) was found to prevent excessive oxidation of ethanol (i.e., ethanol oxidation to C2 not coupled to chain elongation) or oxidation of carboxylates [37,98]. In batch microbiome studies using synthetic C2-rich media, C6 and C8 were only produced in the presence of H_2_, even in the absence of ethanol [70,91]. If *p*H_2_ is too high, carboxylates are reduced to their corresponding alcohol [168]. In addition, a *p*H_2_ above ~0.1 bar reduces the thermodynamic favourability of ethanol oxidation to C2 for ATP generation [58,81]. In general, it is stated that the *p*H_2_ should be above ~0.03 bar to avoid excessive ethanol oxidation to C2 while remaining below ~1.5 bar to prevent carboxylate reduction [106] (Table 5).

Another important headspace component is CO_2_. With a MMC membrane biofilm reactor MCCA were produced solely from a 40/60 ratio of CO_2_ and H_2_ [169]. CO_2_ is also a nutritional requirement for some chain elongating bacteria [132]. In addition, CO_2_ partial pressure influences dissolved carbonate and thus the alkalinity. Experimental studies have shown CO_2_ addition in the headspace improved chain elongation [105], and a combination of CO_2_ and H_2_ in the reactor headspace reduced C3 formation [117]. CO_2_ dosing within the reactor headspace was recently proposed as a key strategy in controlling ethanol-based chain elongation, where high CO_2_ loading rates for ethanol-rich feedstock could stimulate excessive ethanol oxidation to C2, and low values for VFA-rich (and low-ethanol) feedstock could ensure ethanol is used in chain elongation and not for C2 production [98]. Dosing of CO_2_ is inversely proportional to the *p*H_2_, thus care must be taken to ensure minimal *p*H_2_. Weimer et al. [30] calculated the H_2_:CO_2_ ratio that shows the optimal thermodynamics for chain elongation and suggested approx. 1 bar *p*H_2_ with 0.3 bar *p*CO_2_.

Using a complex feedstock will lead to production of CO_2_ and H_2_, since they are both products of primary fermentation by various hydrolytic and acidogenic microorganisms [170]. Therefore, modifying the headspace composition by allowing these gases to accumulate in batch operation in closed vessels [83], working with pressure release in reactors [133] or intermittent opening of reactor headspace for sampling [148], is expected to affect chain elongation. This effect is usually not taken into account in experiments, and to our current knowledge, no studies have specifically assessed the effect of accumulation of these gases in the reactor headspace during primary fermentation on chain elongation. The partial pressure of gases in the reactor headspace should be considered in the design and operation of reactors as plenty of studies demonstrate their influence on chain elongation. However, it must be noted that H_2_ is a highly soluble gas and its concentration in the liquid can be up to 70 times higher than the equilibrium value suggested from mass-transfer coefficients in AD [171]. Therefore, care must be taken to relate *p*H_2_ with H_2_ available for bioconversion processes.

## 9. Reactor Design and the Relation to Retention and Organic Overload

The type of feedstock will influence the choice of fermentation reactor [8]. In the case of a complex, solid-rich feedstock, such as OFMSW or unprocessed food waste, hydrolysis is rate limiting and stirring or pumping is impractical. Therefore, longer retention times are required to allow hydrolysis and solubilisation, and leach bed reactors (LBRs) are an effective option, as they generate a leachate rich in carboxylic acids. For instance, Yesil et al. [172] obtained ± 30 g_COD_ of carboxylic acids per kg of solid waste of which approx. 10% was C6, in batch LBR. Nzeteu et al. [91] used a LBR set-up that allowed semi-continuous operation and obtained a maximum C6 production rate (3.12 g_COD_ L^−1^ d^−1^) from food waste by replacing 75% of the reactor content with fresh feedstock and diluting the leachate with water by 1/15 every 7 days. However, this approach does not easily allow operation with homogenous pH or temperature stability. Another study has shown that operation stability for the production of H_2_ by MMC fermentation of food waste is enhanced by mixing and agitation [116], however, this has not yet been determined for chain elongation.

To exert better process control, complex feedstock can be mechanically pre-treated, i.e., crushing, chopping or blending, to obtain a mixture that can be pumped and stirred, and therefore allows the use of (semi-)continuous stirred tank reactors (sCSTRs). This has been done in some studies focussing on carboxylic acid production from food waste [125,126]. In addition, the feed stream can be diluted with water, or blended with a liquid waste stream or recycled liquor from sludge dewatering to modify the composition. For a more fluid stream such as synthetic feedstock, e.g., ethanol and acetate mixtures, or more easily degradable substrates such as potato-processing or brewery wastewater or food waste leachate, less hydrolysis is required and the chain elongation itself becomes rate-limiting. To allow higher flow rates, whilst maintaining high biomass to counter the rate-liming effect of chain elongation, reactor configurations such as membrane bioreactors [169], CSTRs with in situ settlers [128] and up-flow reactors (UR) with mechanisms for biomass retention, e.g., filter, sludge blankets or packed beds, have been used (Table 3). It has been suggested that biomass retention and cell density are important parameters which are often unreported for MMC fermentation [95]. Indeed, studies using reactors with biomass retention have shown the highest MCCAs production rates so far (Figure 3). Such reactors are well established for other processes, and are worthy of investigation.

In continuous systems, the inoculum acclimation and biomass retention, also expressed as sludge retention time (SRT), favour specific microbial populations. For instance, C4 producers have often been shown to have a longer doubling time than lactic acid producers, and hence wash out more easily from continuous systems [47]. In stirred tank reactors without a settling phase, the SRT is equal to the hydraulic retention time (HRT) and inversely proportional to the OLR. Therefore, in order to operate at a sufficient biomass retention, the OLR can only increase with an increase of substrate COD composition. A suboptimal residence time results in lactic acid or VFA accumulation without chain elongation, and a reduction in hydrolysis [97,128]. In fermentation of salad and vegetable waste in a sCSTR, a retention time of 10 days left a fraction of the substrate unutilized and no C6 was produced; whilst retention time of 20 and 30 days increased C6 production [126]. Similarly, a step-wise decrease of HRT from 20 to 12 to 8 days in cheese whey fermentation, generated C6 concentrations which decreased from 2.24 g_COD_ L^−1^ to 1.45 g_COD_ L^−1^ to 0.41 g_COD_ L^−1^. Analysis of the microbiome composition revealed certain microbial groups were removed by lowering the HRT, via washout, with a dominant presence of lactic acid-producing *Lactobacillus sp*. at HRT of 8 days [123]. Thus, a minimum HRT is required to sustain chain elongation when working with reactors without biomass retention. However, if the HRT is too high competing processes such as methane production are more favoured [36,126]. Methanogens are relatively slow growers, thus reducing HRT has been suggested as a tactic to wash them out, and increase MCCA production rates [72,106]. Therefore, a compromise to reduce methanogens and enhance chain elongation must be found.

Studies on chain elongation have varied HRT from less than 1 day to over 2 weeks resulting in various MCCA production yields (Figure 5). A lower HRT can generate similar yields by altering reactor design to include biomass retention and hence decoupling the HRT from the SRT. Using an UR with biomass retention, chain elongation was performed using a HRT of only 4 h, resulting in a maximum MCCA production rate of 57.4 g L^−1^ d^−1^ using a synthetic ethanol and C2 feed supplemented with methanogenic inhibitors, yeast extract and CO_2_ at neutral pH [105]. In addition, increasing HRT reduces product toxicity by eliminating accumulation, e.g., by gradually reducing HRT from 20 to 2.5 days C6 production from cheese whey rate increased to ~5 g_COD_ L^−1^ d^−1^ [100].

It is important to note that other operational factors come in to play alongside HRT. For instance, increasing HRT from 8 to 12 days and operating at 35 °C increased C6 concentrations in food waste fermentation from approx. 1.64 g_COD_ L^−1^ to 6.55 g_COD_ L^−1^, and even more C6 (10.26 g_COD_ L^−1^) was obtained at a HRT of 8 days by operating at 45 °C [125]. Therefore, the optimal HRT to stimulate chain elongation in complex feedstock fermentations will vary according to the type of system used, as it strongly depends on the reactor configuration, hydrolysis rate, sludge retention time, and other operational parameters such as pH and temperature.

## 10. Overcoming Product Toxicity by In Situ Extraction, Biofilm Formation or Acclimation

Low MCCA concentrations in the fermentation broth result in poor product recovery and high cost of down-stream processing. MCCA concentration can be limited in MMC fermentation for three reasons: (i) the substrate is poor in electron donors or in easily biodegradable organics, and hence prevents in situ substrate accumulation to drive chain elongation; (ii) product accumulation lowers the thermodynamic favourability of chain elongation; and (iii) the antimicrobial properties of MCCAs in their protonated state can result in product toxicity. To evaluate which cause is limiting MCCA production, Weimer et al. [40] measured residual substrate and product concentrations and calculated ΔG for C6 production. Incomplete substrate consumption in their MMC fermentation study enriched with *C. kluyveri* and ethanol-supplemented lignocellulosic feedstock still gave a negative ΔG, thus suggesting product toxicity and/or limited incubation time had prevented further MCCA production [40]. Toxicity limits can be offset by employing in situ extraction, or biofilm formation and/or acclimation of the MMC.

In situ extraction methods have the advantage of continuously removing carboxylic acids from the fermentation broth, thereby alleviating product toxicity and thermodynamic constraints [130]. Various in situ extraction systems for carboxylic acids have been proposed for pure culture and MMC fermentation (Table 6). Electrochemical extraction has been applied to recover C2 to C6 from stillage fermentation, and has shown to simultaneously control pH and stimulate chain elongation by OH^−^ and H_2_ production at the cathode [23]. However, for MMC fermentation of a complex substrate, a MCCA-selective extraction method is preferred to maintain low MCCA concentrations, whilst VFAs remain in the fermentation broth as substrates for chain elongation. Pertraction has been used in various MMC studies as an in situ extraction method selective for MCCA. This is an in-line liquid-liquid, membrane-assisted extraction method driven by a pH-gradient, and is usually performed with mineral oil containing a phase transfer catalyst, e.g., TOPO, and an alkaline recovery phase [37,77,93,96,130]. Pertraction can be combined with membrane electrolysis to drive further separation and obtain a MCCA-rich oil [174]. Whilst the majority of in situ extraction studies report enhanced production rates and chain elongation, two studies did not report a significant improvement [62,155]. The advantages of implementing in situ extraction systems must outweigh the increased complexity and cost in process operation.

Recently, different strategies to overcome product toxicity have been suggested. Allowing biofilms and microscale aggregates to develop improves interactions within a microbiome and tolerance to toxic compounds [175]. Addition of 2 g L^−1^ biochar to a UR fed with synthetic ethanol and C2 significantly improved MCCA production and reduced by-product formation [102]. In this case, microscopic observations revealed the community structure and the spatial distribution of microorganisms changed to dense microbial aggregates around the biochar; it was postulated this improved cell-cell interactions and energy efficiency via stabilising relationships between trophic partners, and increased tolerance to product toxicity [102]. Formation of microbial aggregates has been noted for bioreactors without providing a specific means of biofilm formation; granules were formed in a C3 and ethanol fed CSTR producing C7 as chain elongation product [173]. Therefore, reactor configuration and feedstock that allow microbial aggregates are expected to improve chain elongation, yet research on this is limited. In addition, recent research found the MCCA concentration in the fermentation broth influenced the microbial community structure. It has been suggested that elevated C6 and C8 concentrations lead to a more acclimatized and resistant MMC [92,169]. A microbiome adapted to operating at elevated C6 concentrations had a 10 times higher productivity in an environment with elevated C6 (33 g_COD_ L^−1^) [160]. Further development of a resistant, highly productive MMC would allow accumulation of MCCA resulting in less complex extraction methods with reduced economic burden due to downstream processing.

## 11. MMC Fermentation Scale up and Integration within a Bio-Refinery Context

Biological MCCA production is mostly in the experimental phase, but some scale-up has also been studied. For example, Hegner et al. [101] showed MCCA-producing MMC can be scaled up from 0.11 L serum bottles to 2.2 L bioreactors by maintaining similar reactor operation and without loss of performance or a change in microbial composition. Pilot scale projects are now being started. For instance, the MixAlco^TM^ process has been operated in four parallel 3.78 m^3^ scale fed-batch fermenters processing chicken manure, urea and shredded paper to produce a mixture of carboxylate salts from processed fermentation effluent (containing approx. 6.8 g_COD_ L^−1^ C6) in an 11-month time period as precursors for jet fuel and gasoline [179]. The first start-ups and university spin-offs using MMC fermentation for production of MCCA and other bio-based standard chemicals as starting to appear, such as ChainCraft B.V. in the Netherlands [180].

In order to increase the potential of food waste as a feedstock for production of renewable chemicals, MCCA-producing MMC fermentation can be integrated within a bio-refinery. The term bio-refinery, defined as “the sustainable processing of biomass into a spectrum of marketable products and energy”, is inspired by traditional oil refineries where biomass replaces fossil fuels as feedstock for coproducing chemicals and power through various conversion technologies [181]. Process integration allows production of various compounds such as fuels, chemicals, solvents, biomaterials, food and feed ingredients, fibres and heat and power, thus increasing resilience and robustness against market price fluctuations while minimizing waste [182]. Using bio-waste as the renewable biomass feedstock, known as the 3rd generation bio-refinery concept, not only allows replacement of fossil fuel sources with a renewable alternative, but also stabilises waste streams with maximal use of resource, thus contributing to a circular economy [15]. Integration into a bio-refinery concept could include mechanical pre-treatment of waste streams to obtain pumpable mixtures, or pre-fermentation steps to obtain streams rich in lactic acid or ethanol. The combination of physical and biological processes for organic waste valorisation including MCCA will favour a myriad of product and energy goods that are market competitive (Table 7). Agler et al. gave an overview of bio-, thermo-, or electro-chemical post-processes, to convert carboxylic acids from fermentation into carbonyls, esters, alcohols or alkanes applicable as bulk fuels, solvents [26].

## 12. Conclusions

MCCAs, such as caproic and caprylic acid, are compounds of interest due to their broad range of potential applications. In contrast to chemical or single-culture biotechnological processes, using the consorted action of MMC allows to produce MCCA from complex organic feedstocks, such as food waste, in open, non-sterile systems via the natural process of chain elongation. However, the yields, concentrations and selectivity of this process must be improved in order to increase its viability. Therefore, we have summarised the current knowledge on the underlying mechanism of chain elongation by MMC, discussed the current state of the art on the use of complex organic feedstock and reviewed key operational parameters, and their interactions.

Some of the key findings lie with the fact that with complex substrates and microbial cultures, there must be a greater emphasis on managing competing reactions and positively selecting for chain elongation microbiomes. Since the microbial diversity of MMC ecosystems has been shown to be distinct from pure cultures and clean substrates, existing thermodynamic and kinetic models should be expanded to include complex feedstock and mixed cultures. Advances in microbial culture analysis, such as improved implementation of various “-omics” methods on complex samples, will boost current understanding of MMC fermentation.

Most common complex feedstocks trialled so far include residues from the bio-ethanol and dairy industries, different types of cellulosic wastes, syngas fermentation effluent and different types of organic food waste. These type of feedstocks have resulted in maximum production rates up to 8.02 g_COD_ L^−1^ d^−1^. Supplementation of complex waste-derived feedstock with chain elongation substrates such as ethanol increased production rates, with maxima reported up to 62.8 g_COD_ L^−1^ d^−1^. However, the negative environmental effects from chemical addition have also been reported. The use of synthetic substrates allowed production rates up to 115.2 g_COD_ L^−1^ d^−1^.

Through an extensive review of the literature, including studies targeting MCCAs or reporting MCCAs as by-products, various key operational parameters were identified and discussed to highlight the research gaps. Mesophilic temperatures are so far a preferred choice for chain elongation, yet there is little justification for this. The preferred operational pH seems to lie in a slight acidic range from pH 5 to 7, in order to limit the activity of methanogens. The relationship between organic loading rate (OLR) and MCCA production rates showed a positive correlation to some extent, however this is complicated by the degree of biodegradability of the feedstock. Linked to the organic load is the substrate-inoculum ratio (F/M) at the start-up of the process which favours the accumulation of intermediate compounds instead of methane production when F/M > 5. In addition, whilst increased OLR tends to improve chain elongation, this must be coupled with sufficiently long residence times and biomass retention. OLR and retention times will have to be optimized depending on whether the reactor design has included mechanisms for biomass retention, and the biodegradability of the feedstock.

The literature study revealed very little information is available on some specific operational parameters that have been studied for other MMC applications. For example, in similar MMC fermentation processes a minimal alkalinity was beneficial to stabilise the process and reduce the need for pH controlling agents. However, the buffer capacity required to stimulate chain elongation has not been thoroughly investigated. The partial pressures of CO_2_ and H_2_ in the reactor headspace have been identified to influence chain elongation, however production of these gases during fermentation, and their accumulation in reactor headspace is rarely considered. In order to circumvent the antimicrobial limitations imposed by MCCAs on the microbiome, in situ extraction is often proposed, but the alternative strategies which promote the development of biofilm or granule formation, and MMC adaptation, are worthy of further research. Finally, the development of down-stream processing methods, and integration within a bio-refinery context, are crucial issues to transform MCCA production from organic waste streams into a competitive waste valorisation technology that will contribute to the development of a circular economy.

## Figures and Tables

**Figure 1 molecules-24-00398-f001:**
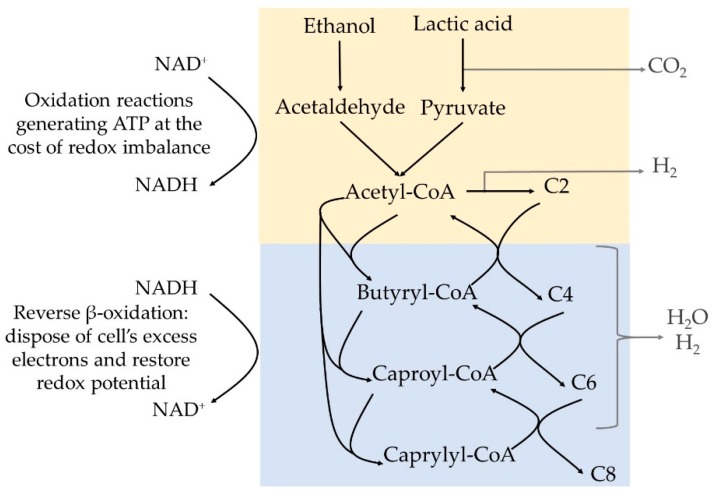
Simplified chain elongation pathways using ethanol or lactic acid as electron donors based on the metabolic pathways described in [29].

**Figure 2 molecules-24-00398-f002:**
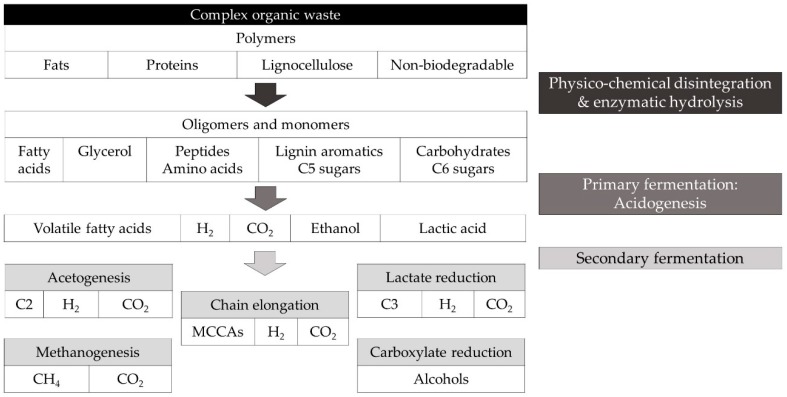
Simplified overview of fermentation pathways that can occur in MMC.

**Figure 3 molecules-24-00398-f003:**
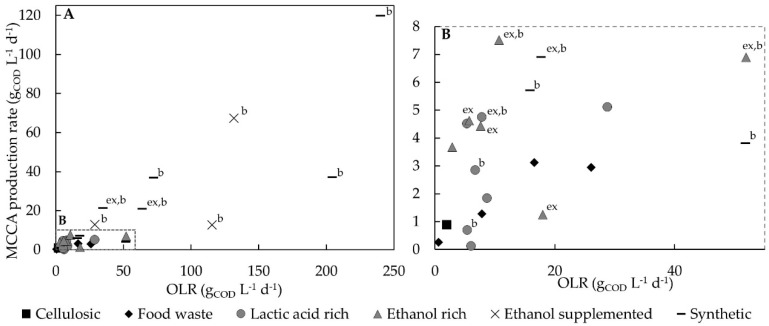
The highest reported MMCA (C6 to C9) production rate as a function of OLR and type of feedstock from 27 different studies using (semi-)continuous MMC fermentation towards MCCAs, VFAs or H_2_. Experiments including in situ product removal or biomass retention are marked with “ex” or “b”, respectively. Studies using complex feedstock are only included which report production rates and OLR in g_COD_ L^−1^, or where such values can be estimated using calculations described in Appendix A. Studies using synthetic substrates only include those listed in Table 3, and three additional studies which used synthetic media comprising ethanol and C2 to represent highest reported rates with HRT optimisation [72,105] and in situ extraction [96] respectively. Data collected from [23,27,36,37,72,77,88,91,92,93,94,95,96,97,100,105,106,110,111,115,116,123,125,126,127,128,154], the full data for the figure can be consulted in the following database [55].

**Figure 4 molecules-24-00398-f004:**
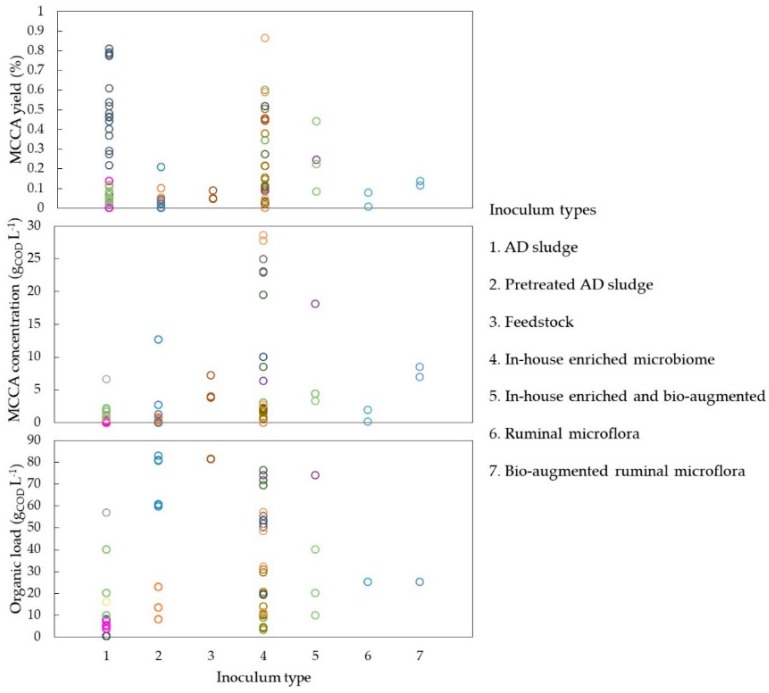
Inoculum type used in 17 studies performing batch experiments of complex or synthetic feedstock in function of the obtained MCCA yield (top), MCCA concentration (middle), or organic load applied (bottom). Colours represent experimental results reported in the same study. Estimated values were calculated as reported in Appendix A. Data collected from [20,40,70,83,88,90,91,96,101,108,122,126,148,157,161,162,163], the full data for the figure can be consulted in the following database [55].

**Figure 5 molecules-24-00398-f005:**
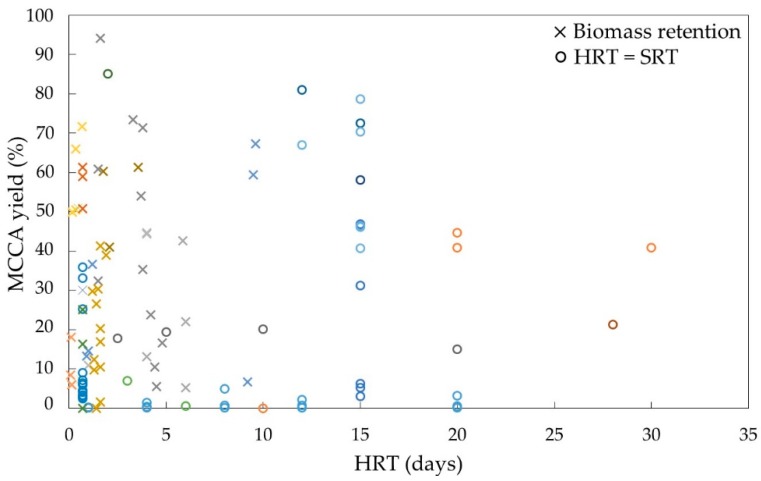
MCCA yield as a function of HRT in 24 studies performing (semi-)continuous experiments of complex or synthetic feedstock. Colours represent experimental results reported within the same study. Estimated values were calculated as reported in Appendix A. Data collected from [23,27,36,37,72,77,88,93,94,95,96,97,98,100,105,106,110,111,115,123,126,127,128,173], the full data for the figure can be consulted in the following database [55].

**Table 1 molecules-24-00398-t001:** Chain elongation reactions via ethanol and lactic acid and thermodynamic information with concentrations and pressures of all components at 1 M or 1 bar, pH 7 at 25 °C.

Equation	Chain Elongation Stoichiometry	ΔG_r_° (kJ mol^−1^)	Ref.
**Via Ethanol-Coupled Reactions: As Determined in *C. kluyveri***			
Overall chain elongation to C4			
1.a	1× Ethanol oxidation	10.50	[29]
	(CH_3_CH_2_OH + H_2_O→CH_3_COO^−^+ H^+^ + 2 H_2_) × 1		
1.b	5× Reverse β-oxidation to C4	−193.00	[29]
	(CH_3_CH_2_OH + CH_3_COO^−^→CH_3_(CH_2_)_2_COO^−^ + H_2_O) × 5		
1.	6 CH_3_CH_2_OH + 4 CH_3_COO^−^→5 CH_3_(CH_2_)_2_COO^−^ + H^+^ + 2 H_2_ + 4 H_2_O	−182.50	[29]
Overall chain elongation to C6			
2.a	1× Ethanol oxidation	10.50	[29]
	(CH_3_CH_2_OH + H_2_O→CH_3_COO^−^ + H^+^ + 2 H_2_) × 1		
2.b	5× Reverse β-oxidation to C6	−194.00	[29]
	(CH_3_CH_2_OH + CH_3_(CH_2_)_2_COO^−^→CH_3_(CH_2_)_4_COO^−^ + H_2_O) × 5		
2.	6 CH_3_CH_2_OH + 5 CH_3_(CH_2_)_2_COO^−^→CH_3_COO^−^ + 5 CH_3_(CH_2_)_4_COO^−^ + H^+^ + 2 H_2_ + 4 H_2_O	−183.50	[29]
**Via Ethanol-Alternative Stoichiometry**			
Overall chain elongation to C6			
3.	12 CH_3_CH_2_OH + 3 CH_3_COO^−^→5 CH_3_(CH_2_)_4_COO^−^ + 4 H_2_ + 8 H_2_O + 2 H^+^	−30.55	[69]
4.	6 CH_3_CH_2_OH + 3 CH_3_COO^−^→ 3 CH_3_(CH_2_)_2_COO^−^ + CH_3_(CH_2_)_4_COO^−^ + 2 H_2_ + 4 H_2_O + H^+^	−183.00	[56]
5.	CH_3_COO^−^ + 2 CH_3_CH_2_OH→CH_3_(CH_2_)_4_COO^−^ + 2 H_2_O	−79.00	[70]
Reverse β-Oxidation to C8			
6.	CH_3_CH_2_OH + CH_3_(CH_2_)_4_COO^−^→CH_3_(CH_2_)_6_COO^−^ + H_2_O	NS	[62]
**Via Lactic Acid**			
Lactic acid to C2 for ATP generation			
7.	CH_3_CH(OH)COO^−^ +H_2_O→CH_3_COO^−^ + 2 H_2_ + CO_2_	−8.79	[69]
Overall Chain Elongation to C4			
8.	CH_3_CH(OH)COO^−^ + CH_3_COO^−^ + H^+^→CH_3_(CH_2_)_2_COO^−^ + H_2_O + CO_2_	−57.52	[69]
Overall chain elongation to C6: as determined for *M. elsdenii*			
9.	CH_3_CH(OH)COO^−^ + CH_3_(CH_2_)_2_COO^−^ + H^+^→CH_3_(CH_2_)_4_COO^−^ + H_2_O + CO_2_	−57.65	[69]

NS, not specified.

**Table 2 molecules-24-00398-t002:** Biochemical reactions that compete with chain elongation and their thermodynamic information with concentrations and pressures of all components at 1 M or 1 bar.

Equation	Competitive Reactions for Chain Elongation	ΔG_r_° or ΔG_r_°’(kJ mol^−1^)	Ref.
10.	Hydrogenotrophic methanogenesis	−125.84 ^a^	[26]
	4 H_2_ + CO_2_→CH_4_ + 2 H_2_O		
11.	Acetoclastic methanogenesis	−39.06 ^a^	[26]
	CH_3_COO^−^ + H^+^→CH_4_ + CO_2_		
12.	Sulphate reduction		[69]
	CH_3_COO^−^ + 3 H^+^ + SO_4_^−2^→H_2_S + 2 H_2_O + 2 CO_2_	−64.39 ^b^	
	2 CH_3_CH(OH)COO^−^ + SO_4_^−2^→H_2_S + 2 CH_3_COO^−^ + 2 HCO_3_^−^	−82.92 ^b^	
	2 CH_3_CH_2_OH + SO_4_^−2^→H_2_S + 2 CH_3_COO^−^ + 2 H_2_O	−69.29 ^b^	
	4 H_2_ + 2 H^+^ + SO_4_^−2^→H_2_S + 4 H_2_O	−39.70 ^b^	
13.	Lactate reduction to C3: as found in *Selenomonas ruminantium*		[26]
	CH_3_CH(OH)COO^−^ + H_2_O→CH_3_COO^−^ + CO_2_ + 2 H_2_ × 1	28.51 ^a^	
	CH_3_CH(OH)COO^−^ + H_2_→CH_3_CH_2_COO^−^ + H_2_O × 2	−86.63 ^a^	
14.	Lactate reduction to C3: as determined for *C. propionicum*	−83.80 ^b^	[47]
	CH_3_CH(OH)COO^−^ + H_2_→CH_3_CH_2_COO^−^ + H_2_O		
15.	Carboxylate to alcohol reduction with H_2_		[12]
	CH_3_COO^−^ + H^+^ + 2 H_2_→CH_3_CH_2_OH + H_2_O	−7.22 ^a^	
	CH_3_CH_2_COO^−^ + H^+^ + 2 H_2_→CH_3_(CH_2_)_2_OH + H_2_O	−7.49 ^a^	
	CH_3_(CH_2_)_2_COO^−^ + H^+^ + 2 H_2_→CH_3_(CH_2_)_3_OH +H_2_O	−3.58 ^a^	
	CH_3_(CH_2_)_4_COO^−^ + H^+^ + 2 H_2_→CH_3_(CH_2_)_5_OH +H_2_O	−7.55 ^a^	
16.	Ethanol oxidation: as determined for *C. formicoaceticum*	−76.90 ^b^	[47]
	2 CH_3_CH_2_OH + 2 CO_2_→3 CH_3_COO^−^ + 3 H^+^		
17.	Coupled ethanol oxidation and C3 reduction		[26]
	CH_3_CH_2_OH + H_2_O→CH_3_CHOO^−^ + H^+^ + 2 H_2_ × 1	7.22 ^a^	
	CH_3_CH_2_COO^−^ + H^+^ + 2 H_2_→CH_3_(CH_2_)_2_OH + H_2_O	−7.49 ^a^	

^a^ at 37 °C, pH 6.82; ^b^ at 25 °C, pH 7.

**Table 3 molecules-24-00398-t003:** Overview of feedstock, operating parameters and product outcome (as C6 production and C7 or C8 when data available) in continuous MMC fermentation.

Feedstock	Reactor (a)	pH	T	OLR	HRT	C6 (C7, C8)		Target Product	Ref.
		-	°C	g_COD_ L^−1^ d^−1^	d	g_COD_ L^−1^ d^−1^	g_COD_ L^−1^		
**Ethanol Rich**									
Diluted yeast fermentation beer	SBR (48 h) ^b^	5.5	30	10.70	15	7.52	NA	MCCAs	[37]
Diluted yeast fermentation beer	SBR (48 h)	5.5	30	5.70	12	4.62	NA	MCCAs	[93]
Diluted wine fermentation residue	UR ^b^	5.2	37	51.90	0.9	4.1 (0.3, 2.5)	NA	MCCAs	[97]
Yeast fermentation beer and thin stillage	SBR (24 h)	5.5	35	2.89	7	2.55	17.90	MCCAs	[92]
Yeast fermentation beer	SBR (48 h)	6.5	30	7.64	15	1.74	NA	MCCAs	[27]
Syngas fermentation effluent, with nutrients	UR ^b^	5.5	30	51.80	0.58	3.80	2.25	MCCAs	[110]
**Lactic Acid Rich**									
Acid whey from quark industry	UR ^b^	5.5	30	28.80	2.5	5.12	NA	MCCAs	[100]
Pre-fermented acid whey yoghurt industry	UR ^b,ex^	5.0	30	10.90	2.1	3.08	<1.00	MCCAs	[94]
Diluted yellow water	USBR (67.2 h)	6.0	30	8.67	28	1.85	51.70	MCCAs	[88]
Diluted cheese whey powder	UR ^b^	6.0	37	5.36	4	0.70	2.80	VFAs	[111]
Cheese whey	SBR (24 h)	NA	35	6.00	12	0.12 *	1.45 *	VFAs	[123]
**(Ligno) Cellulosic Based**									
Switchgrass-derived stillage	CSTR	5.5	35	7.20	2	5.74 (NA, 0.66)	18.70 (NA, 2.40)	MCCAs	[99]
Pre-fermented grass	SBR (24 h)	NA	32	5.30	2	4.52	9.03	MCCAs	[95]
Corn-derived thin stillage	SBR (48 h) ^ex^	5.4-5.7	35	18.00	3	1.14 (0.10, 0.00)	NA	MCCAs	[23]
Pre-treated corn fibre: dilute-acid/dilute-alkali/hot water	SBR (48 h)	5.5	55	1.92	15	0.6 */0.39 */0.12 *	NA	VFAs	[36]
Paper fines and industrial bio-sludge (40:60)	SBR ^a^	NA	40	2.6 g_VS_ L^−1^	16	0.42 (0.08, NA)	6.67 (1.36, NA)	MCCAs	[124]
**Food Waste**									
Simulated food waste	SBR (24 h)	5-6	34	30 g_TS_ L^−1^	3	8.09 *	24.30 *	H_2_	[116]
Restaurant food waste	LBR (7 days)	NA	37	16.60	7	3.12	21.80	MCCAs	[91]
Cafeteria food waste	SBR (24 h)	5.5	45	9 g_TS_ L^−1^ d^-1^	8	1.28	10.30	VFAs	[125]
Vegetable and salad waste	SBR (24 h)	5-7.5	35	0.57	20	0.25 *	5.08 *	VFAs	[126]
**Other**									
Sucrose-rich synthetic wastewater	UR	3.6	35	15.80	0.71	5.70	4.03	H_2_	[115]
Synthetic glucose medium	UR ^b^	NA	30	204.00	0.10	4.13 *	0.43 *	H_2_	[127]
Primary and waste activated sludge	SBR (24 h)	NA	35	79.90	1	< 0.08 *	< 0.08 *	VFAs	[123]
**Food Waste with Ethanol Supplementation**									
Pre-fermented OFMSW, 44.8 g_COD_ L^−1^ ethanol, CO_2_	UR ^b^	6.75	30	NA	0.46	60.70 (4.59, 2.13)	27.80 (2.10, 0.98)	MCCAs	[106]
Pre-fermented food waste, 78.4 g_COD_ L^−1^ ethanol, CO_2_	CSTR ^b^	6.8	30	28.90	4	12.80	51.20	MCCAs	[128]
Pre-fermented food waste, 78.4 g_COD_ L^−1^ ethanol, CO_2_	CSTR ^b^	6.8	30	115.00	1	12.40 (0.33, 0.42)	15.70 (0.18, 0.18)	MCCAs	[128]

SBR: sequential batch reactor; UR: up-flow reactor; CSTR: continuous stirred tank reactor; NA: not applicable; ^a^ fermentation cycle in case of SBR or semi-batch; ^b^ biomass retention; ^ex^ in situ extraction; * estimated from graphs. When values were not directly reported by cited study these were estimated as described in Appendix A.

**Table 4 molecules-24-00398-t004:** Reported environmental conditions to select for chain elongation in MMC fermentation for MCCA production.

	Summarised Literature Findings	Ref.
**Temperature**		
Preference	Mesophilic range from 30 to 45 °C	[49,125,129,130,131]
Effect	Indications of influence on microbiome composition.	[129]
**pH**		
Preference	Slight acidic: preferred range from pH 5 to 6	[100,123,130,134]
	Neutral pH, if specific methanogen inhibitors added	[105,106]
Self-regulation	Without control, pH usually stabilises between 5.5 and 6.7	[90,95,135]
Product toxicity	Protonated C6 toxicity limit: 1.92 g_COD_ L^−1^ C6 accumulation over toxicity limits is possible (in batch, one point measurement)	[37,58,95]
**Alkalinity**		
MCCA	No data available	
AD	1 to 5 gCaCO_3_ L^−1^	[144,145]
H_2_	0.11 gCaCO_3_ L^−1^	[146]
VFA	2.4 to above 30 gCaCO_3_ L^−1^	[122,147,148]

**Table 5 molecules-24-00398-t005:** Experimental and theoretical reported values of *p*H_2_ and/or *p*CO_2_ in reactor headspace to stimulate chain elongation in MMC fermentation.

Headspace Requirement	Influence on Chain Elongation	Ref.
**Hydrogen**		
*p*H_2_ < 0.1 bar	Theoretical maximum *p*H_2_ for oxidation of ethanol to C2 for ATP generation to be thermodynamically feasible	[58]
*p*H_2_ < 1.5 bar	Experimental *p*H_2_ minimum to trigger reduction of C2, C3 and C4 to alcohols (pH = 5)	[168]
*p*H_2_ = 1.5 bar	Initiated production of C6 from ethanol and C2 in experiment	[70]
*p*H_2_ > 2.52 × 10^−6^ bar	Theoretical minimum *p*H_2_ to prevent oxidation of C6 at experimental concentrations	[37]
0.03 bar < *p*H_2_ < 1.5 bar	Recommended *p*H_2_ for chain elongation	[106]
**Carbon Dioxide**		
CO_2_ or CO_3_^2−^ addition	Lowered pH, improved MCCAs production	[105]
**Mixture of Hydrogen and Carbon Dioxide**		
*p*H_2_ ≈ 0.007% by dosing CO_2_	Excess oxidation is thermodynamically prevented	[98]
H_2_/CO_2_ = 60/40	MCCAs produced from H_2_/CO_2_ mixture	[169]
H_2_/CO_2_ = 80/20 at 0.5 bar	Enhanced production of C6 from ethanol and C2 in experiment	[91]
*p*H_2_ = 2 bar, *p*CO_2_ = 2 bar	Reduced C3 formation	[117]
*p*H_2_ = 1 bar, *p*CO_2_ = 0.3 bar	Optimal thermodynamics for chain elongation	[30]

**Table 6 molecules-24-00398-t006:** Overview of the results of applying in situ extraction in MCCA fermentation experiments.

In Situ Extraction and Fermentation Method	Results	Ref.
**Biphasic Extractive Fermentation**		
*M. elsdenii* strain, sucrose substrate	58.5 g_COD_ L^−1^ C6 in solvent	[176]
*Clostridium sp.* BS-1, galactitol substrate	70.6 g_COD_ L^−1^ C6 in solvent	[177]
**Anion Exchange Resin**		
*M. elsdenii* strain, glucose substrate	Extraction doubled C6 to 24.3 g_COD_ L^−1^	[178]
**Integrated Cross-Flow Nanofiltration**		
MMC, pre-treated cellulosic feedstock	7× lower carboxylic acid concentration in fermenter, no significant yield improvement	[155]
**Membrane Electrolysis**		
MMC, thin stillage feedstock	Lower need for caustic soda addition, C4-C6 from 46% to 70%, cathodic H_2_ formation	[23]
**Pertraction**		
MMC, diluted yeast fermentation beer	MCCA-specific, 4× increase in C6 specificity to 32%, elongation to C8, extraction optimization	[37,93,130]
MMC, Synthetic substrate with C2 and ethanol	4× increase C8 productivity to 0.8 g_COD_ L^−1^ d^−1^	[96]
MMC, Synthetic substrate with C4 and lactic acid	Productivity from 0.6 to 1 g_COD_ L^−1^ d^−1^ C6 by implementing extraction	[77]
*C. kluyveri*, synthetic substrate with C2 and ethanol	No significant difference in production rates	[62]
**Pertraction and Membrane Electrolysis**		
MMC, diluted fermentation beer	Lower need for caustic soda addition, MCCA recovery of 87%, obtained >90% MCCA oil	[174]

**Table 7 molecules-24-00398-t007:** Processes and applications for carboxylic acid-rich (MCCAs and VFAs) fermentation effluent currently described in literature.

Process/Application	Product	Ref.
MixAlco process	Alcohol fuel	[28]
Secondary fermentation	Lipid/biodiesel	[183]
Secondary fermentation	Polyhydroxyalkanoates (PHA)	[147,154,184]
Microbial fuel cell feedstock	Electricity	[185]
Carbon source for bioremediation	Dechlorination of Chloroethenes	[186]
Extraction and Kolbe electrolysis	Liquid alkane fuels, i.e., C10-C20 hydrocarbons	[14,27,95]

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
