# Peer review of "Medium Chain Carboxylic Acids from Complex Organic Feedstocks by Mixed Culture Fermentation"

_molecules, 2019, doi:10.3390/molecules24030398_

Round 1
Reviewer 1 Report
The paper has assessed the literature about volatile fatty acids (VFA) production with anaerobic digestion mixed culture. Volatile acids production using mixed cultures is a fascinating topic. Although this approach is not new (see Zverlov et al. Applied Microbiology and Biotechnology 71(5): 587-597, 2006), there are a lot of gaps in fundamental knowledge to improve process stability and yielding of both acids and hydrogen production.
The review is very comprehensive and put together relevant and new literature about VFA production. But, I feel that the authors missed an important research approach which comprises the physicochemical pretreatment of the inoculum (manly from anaerobic digestion). Behind this idea is to offer a selective pressure to the microbial communities, thus inhibiting (or getting rid of) methanogenic consortia to improve both VFA and hydrogen production. This approach could be very interesting, once the authors had stated in this draft that bio-augmentation did not enhance the yielding of VFA production, at least not significantly.
I want to suggest the authors two papers that I find useful and addresses this approach, but aiming hydrogen production:
Ren et al. International Journal of Hydrogen Energy 33(16): 4318-4324, 2008 and Pendyala et al. International Journal of Hydrogen Energy 37(17): 12175-12186, 2012. But there are many others articles about this line of research, so I would suggest to looking towards this direction.
Another set of papers that might be crucial for this review is the work of Giovannini et al. International Journal of Hydrogen Energy 41: 17713-17722 (2016) and Harper & Pohland Biotechnology and Bioengineering 28(4):585-602 (1986). Both papers discuss the role of hydrogen partial pressure in the thermodynamics of anaerobic digestion, and how it controls the type of VFA (butyric, acetic or propionic acids) yielding. Finally, I suggest the work of Pauss et al. Applied Environmental Microbiology 56(6):1636-1644 (1990), which discusses that in anaerobic digestion, the hydrogen concentration in the liquid phase could be up to 70 times the equilibrium values. Thus the measurement of hydrogen amount in biogas could be useless to predict the VFA yielding in anaerobic digestion.
Author Response
Response to reviewer 1 comments on the manuscript “Medium chain carboxylic acids from complex organic feedstock by mixed culture fermentation”
Point 1: The paper has assessed the literature about volatile fatty acids (VFA) production with anaerobic digestion mixed culture. Volatile acids production using mixed cultures is a fascinating topic. Although this approach is not new (see Zverlov et al. Applied Microbiology and Biotechnology 71(5): 587-597, 2006), there are a lot of gaps in fundamental knowledge to improve process stability and yielding of both acids and hydrogen production. The review is very comprehensive and put together relevant and new literature about VFA production.
Response 1: The positive comments regarding the topic and overall review are much appreciated by the authors and we thank the reviewer for their time and consideration. As the reviewer mentioned, bioproduction from waste has been studied for years, for example to produce acetone-butanol (Zverlov et al. 2006) or other soluble compounds such as VFA (e.g. as reviewed by Bolzonella et al. Ind. Eng. Chem. Res. 44(10): 3412-3418, 2005). However, we would like to point out that our review focusses specifically on medium chain carboxylic acids (MCCAs), which are a product from VFA and alcohols or lactic acid, and the operational conditions for this process, which are still largely unknown. Thus, we believe this work would fill some gaps in this particular field. Our response and corrections to further comments are below.
Point 2: But, I feel that the authors missed an important research approach which comprises the physicochemical pretreatment of the inoculum (manly from anaerobic digestion). Behind this idea is to offer a selective pressure to the microbial communities, thus inhibiting (or getting rid of) methanogenic consortia to improve both VFA and hydrogen production. This approach could be very interesting, once the authors had stated in this draft that bio-augmentation did not enhance the yielding of VFA production, at least not significantly. I want to suggest the authors two papers that I find useful and addresses this approach, but aiming hydrogen production: Ren et al. International Journal of Hydrogen Energy 33(16): 4318-4324, 2008 and Pendyala et al. International Journal of Hydrogen Energy 37(17): 12175-12186, 2012. But there are many others articles about this line of research, so I would suggest to looking towards this direction.
Response 2: We thank the reviewer for suggesting to include this. The pretreatment of inoculum has indeed been studied for VFA, hydrogen and similar mixed culture fermentation products, yet we found limited information on this specifically applicable to chain elongation and it was therefore not included. However, we have re-analysed the literature while focussing on inoculum pretreatment and made changes accordingly to what we found.
First, we found one of the 16 studies included in Figure 4 had pretreated the AD sludge-derived inoculum. We found another study that included pretreated AD sludge that focussed on VFA production, but also showed some production of caproic acid (C6), and we added it to Figure 4. Therefore, we included a separate inoculum category for “pretreated inocula” in Figure 4 (Page 20, in the manuscript in track changes). Secondly, we inserted lines 533-543 (Page 18, in the manuscript in track changes) to explain how pretreatment of inoculum has been found to be important for mixed culture fermentation and limited knowledge is available for MCCA production specifically. We have included the suggested papers focussing on hydrogen, and added a few other VFA-aimed papers to complement the discussion.
Point 3: Another set of papers that might be crucial for this review is the work of Giovannini et al. International Journal of Hydrogen Energy 41: 17713-17722 (2016) and Harper & Pohland Biotechnology and Bioengineering 28(4):585-602 (1986). Both papers discuss the role of hydrogen partial pressure in the thermodynamics of anaerobic digestion, and how it controls the type of VFA (butyric, acetic or propionic acids) yielding.
Response 3: We have added a reference to both papers (Section 8.2, Line 595-598, Page 21 of the manuscript with track changes) as we agree that it fits well within the review. We would like to thank the reviewer to suggest the work of Giovannini et al. as it served as inspiration for the addition of Table 5 to provide a clearer overview of the presented data.
Point 4: Finally, I suggest the work of Pauss et al. Applied Environmental Microbiology 56(6):1636-1644 (1990), which discusses that in anaerobic digestion, the hydrogen concentration in the liquid phase could be up to 70 times the equilibrium values. Thus the measurement of hydrogen amount in biogas could be useless to predict the VFA yielding in anaerobic digestion.
Response 4: Indeed, we agree that care should be taken when directly relating partial pressures of gases to their concentration in the fermentation broth. The paper suggested by the reviewer explains this issue well, thus we have added it to Section 8.2 (Lines 631-633, Page 21 of the manuscript with track changes) so readers that are more interested in this particular topic can find the further information.
Reviewer 2 Report
General comment:
The manuscript deals with an overview on the production of medium chain carboxylic acids through fermentation, by mixed culture, of complex organic feedstock.
The manuscript is suitable to be published in this journal; however, some major points should be addressed before publication.
Some minor language mistakes are present that should anyway be corrected.
1. Introduction
Please, improve the introduction by including a short overview on processes for converting waste into energy, chemicals or materials. Please, consider the following papers:
o Molino, A., Larocca, V., Chianese, S., Musmarra, D., 2018, Biofuels production by biomass gasification: A review, Energies 11(4), Article number 811.
o Kumar, V., Longhurst, P., 2018, Recycling of food waste into chemical building blocks, Current Opinion in Green and Sustainable Chemistry, 13, 118-122.
o Chianese, S., Fail, S., Binder, M., Rauch, R., Hofbauer, H., Molino, A., Blasi, A., Musmarra, D., 2016, Experimental investigations of hydrogen production from CO catalytic conversion of tar rich syngas by biomass gasification, Catalysis Today, 277, 182-191.
o Reddy, M.V., ElMekawy, A., Pant, D., 2018, Bioelectrochemical synthesis of caproate through chain elongation as a complementary technology to anaerobic digestion, Biofuels, Bioproducts and Biorefining, 12(6), 966-977.
o Ma, H., Guo, Y., Qin, Y., Li, Y.-Y., 2018, Nutrient recovery technologies integrated with energy recovery by waste biomass anaerobic digestion, Bioresource Technology, 269, 520-531.
2. Chain elongation behaviour of pure cultures can be extended for MMC
Please, compare the performances of bacteria in terms of MCCAs production (I suggest to collect data in a Table with references).
Please, highlight operating conditions used for each bacterium (I suggest to collect data in a Table with references).
5. Environmental conditions that influence chain elongation
Please, summarize data in Tables.
Among the parameters assessed, I would like to suggest to consider the amount of organic wastes produced.
8. The influence of partial pressures in reactor headspace
Please, summarize data in a Table.
11. MMC fermentation scale up and integration within a bio-refinery context
Please, improve this section by including an overview of operating industrial scale plants, detailing feedstocks and operating conditions.
Author Response
Response to reviewer 2 comments on the manuscript “Medium chain carboxylic acids from complex organic feedstock by mixed culture fermentation”
Point 1: General Comments
The manuscript deals with an overview on the production of medium chain carboxylic acids through fermentation, by mixed culture, of complex organic feedstock. The manuscript is suitable to be published in this journal; however, some major points should be addressed before publication. Some minor language mistakes are present that should anyway be corrected.
Response 1: We thank the reviewer for their thorough assessment of our work and the helpful feedback. The manuscript has been checked by a native English speaker, so language mistakes should now be minimized. Our response and corrections to further comments are below.
Point 2: 1. Introduction
Please, improve the introduction by including a short overview on processes for converting waste into energy, chemicals or materials. Please, consider the following papers:
• Molino, A., Larocca, V., Chianese, S., Musmarra, D., 2018, Biofuels production by biomass gasification: A review, Energies 11(4), Article number 811.
• Kumar, V., Longhurst, P., 2018, Recycling of food waste into chemical building blocks, Current Opinion in Green and Sustainable Chemistry, 13, 118-122.
• Chianese, S., Fail, S., Binder, M., Rauch, R., Hofbauer, H., Molino, A., Blasi, A., Musmarra, D., 2016, Experimental investigations of hydrogen production from CO catalytic conversion of tar rich syngas by biomass gasification, Catalysis Today, 277, 182-191.
• Reddy, M.V., ElMekawy, A., Pant, D., 2018, Bioelectrochemical synthesis of caproate through chain elongation as a complementary technology to anaerobic digestion, Biofuels, Bioproducts and Biorefining, 12(6), 966-977.
• Ma, H., Guo, Y., Qin, Y., Li, Y.-Y., 2018, Nutrient recovery technologies integrated with energy recovery by waste biomass anaerobic digestion, Bioresource Technology, 269, 520-531.
Response 2: We thank the reviewer for suggesting these additional resources. The introduction has been extended to include a brief overview on conversion technologies of waste to energy, chemicals or materials, and types of biowaste classification (Lines 37-48 in the manuscript in track changes). Some of the papers suggested by the reviewer have been included, and additional papers were inserted to complement the discussion. We have kept the added text concise, and have provided references to the relevant literature.
In particular, we would like to thank the reviewer for suggesting the paper “Reddy, M.V., ElMekawy, A., Pant, D., 2018, Biofuels, Bioproducts and Biorefining, 12(6), 966-977” as this is an interesting review regarding bioelectrochemical systems for MCCA production that we felt was crucial to include. This paper is now referenced in Lines 132-133 in the manuscript with track changes.
Point 3: 2. Chain elongation behaviour of pure cultures can be extended for MMC
Please, compare the performances of bacteria in terms of MCCAs production (I suggest to collect data in a Table with references).
Please, highlight operating conditions used for each bacterium (I suggest to collect data in a Table with references).
Response 3: We agree that it would be interesting to accumulate this information. The bacteria known for chain elongation are briefly presented in Section 2 in order to explain how much is known about their underlying metabolism since the biochemical reactions derived from pure culture fermentation are key in understanding following sections of the review. However, we wanted to focus this review specifically towards mixed microbial culture fermentation and their operational conditions when using complex feedstock. Therefore, we feel that a detailed table regarding the performance of pure culture studies, which typically use clean and simple substrates, would fall outside the scope of this review. Nevertheless, in order to address the reviewer’s point, we have now included a sentence directing interested readers to appropriate further reviews (Lines 170-173 in the manuscript with track changes).
Point 4: 5. Environmental conditions that influence chain elongation
Please, summarize data in Tables.
Among the parameters assessed, I would like to suggest to consider the amount of organic wastes produced.
Response 4: We thank the reviewer for this valuable suggestion and have now summarized the data in Table 4 to give a clearer overview of the data discussed (page 14 in manuscript with track changes) and to summarize the concluding remarks in the text. In regards to the comments on the amount of organic waste produced, whilst we do recognise that would be an interesting discussion when it comes to the scale-up and industrial implementation of the process, our aim with this section is to focus on the direct effect of environmental conditions. Hence, we have opted for not including this to maintain our original scope of the review.
Point 5: 8. The influence of partial pressures in reactor headspace
Please, summarize data in a Table.
Response 5: As in point 4, we have now added Table 5 (page 22 in manuscript with track changes) as an overview of experimental and theoretical values referenced and discussed in the text.
Point 6: 11. MMC fermentation scale up and integration within a bio-refinery context
Please, improve this section by including an overview of operating industrial scale plants, detailing feedstocks and operating conditions.
Response 6:
We are currently only aware of two operational pilot-scale plants running for production of medium chain carboxylic acids. In addition, limited information is available regarding their operating conditions. Instead, after careful consideration, we propose to complete this section with Table 7, where we summarize current applications for chain elongation fermentation effluent (Page 26 in the manuscript with track changes).
Reviewer 3 Report
After reading the manuscript “Medium chain carboxylic acids from complex organic feedstock by mixed culture fermentation” I consider that the work presented is of good quality. An interesting and detailed review regarding the production of medium chain carboxylic acids from different feedstock considering several fermentation parameters has been performed. However, before the full acceptance of this work for publication I have the following minor observations:
1.- Pg 4, L157: check spelling sterilisation or sterilazation
2.- Pg 7, L243: Correct “thse”
3.- Pg12, L380: Please check sentence "but it also selects for shorter "
4.- Pg13, L405: Check format for reference, e.g. Angenent et al. [38] not Angenent et al.
5.- Pg13, L406: Khor, Check reference style.
6.- Pg14, L441: Check spelling for “stabilisation”.
7.- Pg, 15: Figure 3 it is difficult to follow in the present form, due to the references seem to overlap with symbols. Consider to change the format or better ad this data in a Table.
8.- Pg 17, L571: Weimer, Reference style. Same for L587 “Yesil” and L588 “Nzeteu”.
Author Response
Response to reviewer 3 comments on the manuscript “Medium chain carboxylic acids from complex organic feedstock by mixed culture fermentation”
Point 1:
After reading the manuscript “Medium chain carboxylic acids from complex organic feedstock by mixed culture fermentation” I consider that the work presented is of good quality. An interesting and detailed review regarding the production of medium chain carboxylic acids from different feedstock considering several fermentation parameters has been performed. However, before the full acceptance of this work for publication I have the following minor observations:
Response 1: We are very grateful for the positive comments and acknowledgement of the work done for this review. We would like to thank the reviewer for their thorough assessment of our work. Our response and corrections to further comments is below.
Point 2:
1.- Pg 4, L157: check spelling sterilisation or sterilazation
Response 2: We have checked the spelling, and since we have written the manuscript following the British English spelling, we will keep the word as “sterilisation”. We have double-checked the manuscript to assure British English spelling rules are followed throughout.
Point 3:
2.- Pg 7, L243: Correct “thse”
Response 3: Was corrected to “these” as can be seen in Pg 7, L 269 in the manuscript with track changes. In addition, the document was checked again on typing mistakes and were corrected throughout.
Point 4:
3.- Pg12, L380: Please check sentence "but it also selects for shorter "
Response 4: We have re-written the sentence to improve clarity. Changes can be seen in Pg 13 L408-410 in the manuscript with track changes.
Point 5:
4.- Pg13, L405: Check format for reference, e.g. Angenent et al. [38] not Angenent et al.
Response 5: Reference style has been adjusted as suggested. Changes can be seen in Pg 14 L435 in the manuscript with track changes.
Point 6:
5.- Pg13, L406: Khor, Check reference style.
Response 6: Reference style has been adjusted as suggested. Changes can be seen in Pg 14 L436 in the manuscript with track changes.
Point 7:
6.- Pg14, L441: Check spelling for “stabilisation”.
Response 7: We have now consistently used the British English spelling as explained in Response to point 2
Point 8:
Pg, 15: Figure 3 it is difficult to follow in the present form, due to the references seem to overlap with symbols. Consider to change the format or better ad this data in a Table.
Response 8: We agree it might be a bit confusing to identify the symbols and have a clear overview of the graph due to the references. Therefore, references were removed and added in caption instead, similar as for Figures 4 and 5. In addition, the spreadsheet used to compile the data from the reviewed studies and their processing to create the tables and figures is made available online and referred to (Reference [55] mentioned in L 145-146 of the manuscript with track changes, and referenced at Figure 3 Pg 17, Figure 4 Pg 20 and Figure 5 Pg 24 in the manuscript with track changes). This dataset will be made accessible at point of publication.
55. De Groof, V.; Coma, M.; Arnot, T.; Leak, D.; Lanham, A. Dataset on experimental data available in the literature on "Medium chain carboxylic acids from complex organic feedstock by mixed culture fermentation". University of Bath Research Data Archive, 2019; https://doi.org/10.15125/BATH-00584.
Point 9:
Pg 17, L571: Weimer, Reference style. Same for L587 “Yesil” and L588 “Nzeteu”.
Response 9: The reference style has been adjusted as suggested. Changes can be seen in Pg 24 L712, Pg 22 L 642-643 in the manuscript with track changes.
In addition, the reference style was also corrected accordingly for L 225 “Kucek”, L 246 “Strauber”, L258 “Scarborough”, L 297 “Grootscholten”, L 302-304 “Grootscholten”,and L 756 “ Hegner” in the manuscript with track changes.
Round 2
Reviewer 2 Report
The authors revised the manuscript according to the comments/changes suggested and a general improvement of the manuscript can be found. The review is suitable to be published in this journal in the current form.